# DeepPersona: A Generative Engine for Scaling Deep Synthetic Personas

## Abstract

Simulating human profiles by instilling personas into large language models (LLMs) is rapidly transforming research in agentic behavioral simulation, LLM personalization, human-AI alignment, etc. However, most existing synthetic personas remain shallow and simplistic, capturing minimal attributes and failing to reflect the rich complexity and diversity of real human identities. We introduce DeepPersona, a scalable generative engine for synthesizing narrative-complete synthetic personas through a two-stage, taxonomy-guided method. First, we algorithmically construct the largest-ever human-attribute taxonomy, comprising over hundreds of hierarchically-organized attributes, by mining thousands of real user-ChatGPT conversations. Second, we progressively sample attributes from this taxonomy, conditionally generating coherent and realistic personas, averaging hundreds of structured attributes and roughly 1 MB of narrative text, two orders of magnitude deeper than prior works. Intrinsic evaluations confirm significant improvements in attribute diversity (32% higher coverage) and profile uniqueness (44% greater) compared to state-of-the-art baselines. Extrinsically, our personas enhance GPT-4.1-mini's personalized Q&A accuracy by 11.6% average on ten metrics, and substantially narrow (by 31.7%) the gap between simulated LLM "citizens" and authentic human responses in social surveys. Our generated "national citizens" reduced the performance gap on the Big Five personality test by 17% relative to LLM-simulated citizens. DeepPersona thus provides a rigorous, scalable, and privacy-free platform for high-fidelity human simulation and personalized AI research.

## 1 Introduction

Generating synthetic personas via large language models (LLMs) has rapidly gained popularity, powering applications across personalized assistance Yuan et al. (2023), social and behavioral simulations Lu et al. (2025), interactive role-playing agents Qiu & Lan (2024), and alignment research Castricato et al. (2025). The flexibility and generative power of modern LLMs allow researchers to effortlessly produce large volumes of synthetic human-like profiles, enabling studies and experiments otherwise limited by data scarcity or privacy concerns.

Despite widespread adoption, current synthetic personas often remain shallow and simplistic, failing to capture the depth, diversity, and realism of actual human profiles Ge et al. (2024). Existing approaches typically rely on a handful of manually-defined traits or brief, templated descriptions, which fundamentally limit their complexity Wang et al. (2025b). Moreover, naively using Large Language Models (LLMs) to expand upon seed attributes is fraught with substantial limitations: the resulting narratives frequently lack genuine diversity, exhibit stereotypical or overly optimistic portrayals inherited from training data, and fail to capture the semantic richness and nuanced complexity observed in real individuals Li et al. (2025); Wang et al. (2024a).

To bridge this critical gap, it is necessary to establish rigorous methods capable of systematically scaling synthetic user profiles. An ideal profile generation approach should satisfy several key desiderata. Specifically, it must: (1) scale the coverage of the broad spectrum of real-world human attributes, from demographics to life experiences; (2) scale diversity to capture nuanced, non-stereotypical variations among individuals; and (3) maintain rigorous internal consistency and narrative coherence, while remaining customizable for specific user cohorts or application domains. However, existing methodologies rarely satisfy these requirements simultaneously, revealing a fundamental gap in the scalable generation of deep synthetic personas.

To address these challenges, we introduce DEEPPERSONA, a novel two-stage generative engine to synthesize detailed, diverse, and customizable synthetic user personas. In the 1st stage, we construct a comprehensive human attribute taxonomy by mining thousands of real-world multi-turn conversations from user-ChatGPT interactions. Leveraging natural questions that elicit extensive human self-disclosure, we algorithmically extract and merge attribute phrases into a unified hierarchical structure, resulting in a taxonomy with 8000+ human attribute nodes–far exceeding prior manually-curated persona datasets Hasenfeld (2010). In the 2nd stage, we introduce a progressive attribute sampling algorithm: starting from customizable anchor traits, our method iteratively selects informative attributes conditioned on the existing persona context, incrementally building profiles that maintain internal consistency and narrative realism. This structured, iterative approach enables researchers to precisely control persona generation, systematically explore the space of human attributes, and generate profiles at depth and scale unattainable by naïve LLM sampling Wang et al. (2025b).

We evaluate DEEPPERSONA intrinsically and extrinsically. Intrinsically, we assess attribute coverage, uniqueness, and actionability, showing substantial gains over state-of-the-art persona resources such as PersonaHub Ge et al. (2024) and OpenCharacter Wang et al. (2025b). Extrinsically, we test DEEPPERSONA in two downstream tasks: (1) personalized prompting, where conditioning GPT models Achiam et al. (2023) on deeper personas yields up to 11.6% higher response accuracy; and (2) human-population simulation, where synthetic populations answer World Values Survey questions Tao et al. (2024), reducing deviation from real responses by 31.7%, outperforming strong baselines. (3)In the Big Five personality test, our generated "national citizens" reduced the deviation from ground-truth data by 17% compared to LLM-simulated citizens.

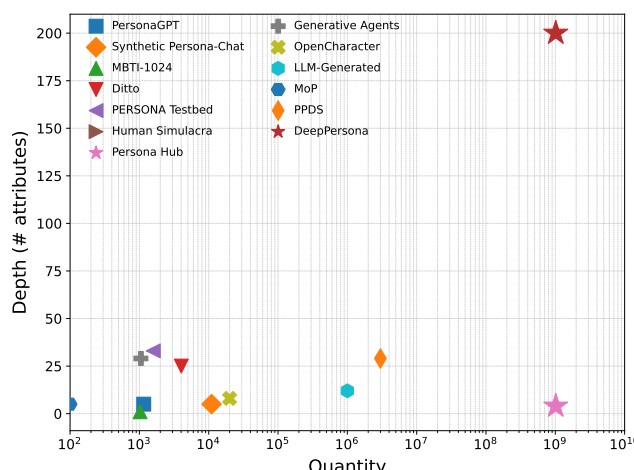

Figure 1: Current persona generation methods face a trade-off between quantity and depth. While approaches like PersonaHub Ge et al. (2024) achieve massive scale with shallow depth, DEEPPERSONA uniquely scales both, automatically enriching PersonaHub's billion profiles with hundreds of structured attributes.

These results demonstrate that DEEPPERSONA synthesizes realistic human identities, enabling scalable, privacy-preserving, and high-fidelity user modeling.

## 2 RELATED WORK

**Synthetic Persona Generation.** Early persona-conditioned dialogue models represented users as short descriptive statements, often limited to a few manually-crafted attributes Zhang et al. (2018). The advent of Large Language Models (LLMs) enabled synthetic persona generation at unprecedented scale: PersonaHub Ge et al. (2024) utilized GPT-4 to produce over one billion brief, attribute-sparse personas, emphasizing quantity rather than semantic depth. OpenCharacter Wang et al. (2025b) extended this by pairing short GPT-generated personas with style-tuned dialogues, enhancing interaction fidelity yet maintaining limited persona depth. Recent intrinsic analyses highlight pervasive issues across these methods, such as insufficient lexical diversity, positivity biases, and demographic under-representation Li et al. (2025). In contrast, DEEPPERSONA systematically addresses these limitations through a taxonomy-guided sampling strategy, enhancing persona depth.

**LLM Personalization.** Personalization in Large Language Models (LLMs) aims to tailor model outputs to individual user identities, preferences, or interaction histories. Prominent approaches include retrieval-augmented prompting Jiang et al. (2025), parameter-efficient user embedding fine-tuning Wang et al. (2024b); Braga (2024), and hybrid architectures integrating external user memory stores. A fundamental bottleneck across these strategies is the superficial nature of existing persona representations, typically limited to brief, shallow attribute sets Wang et al. (2024b); Li et al. (2025). By contrast, DEEPPERSONA generates personas with orders-of-magnitude greater coverage,

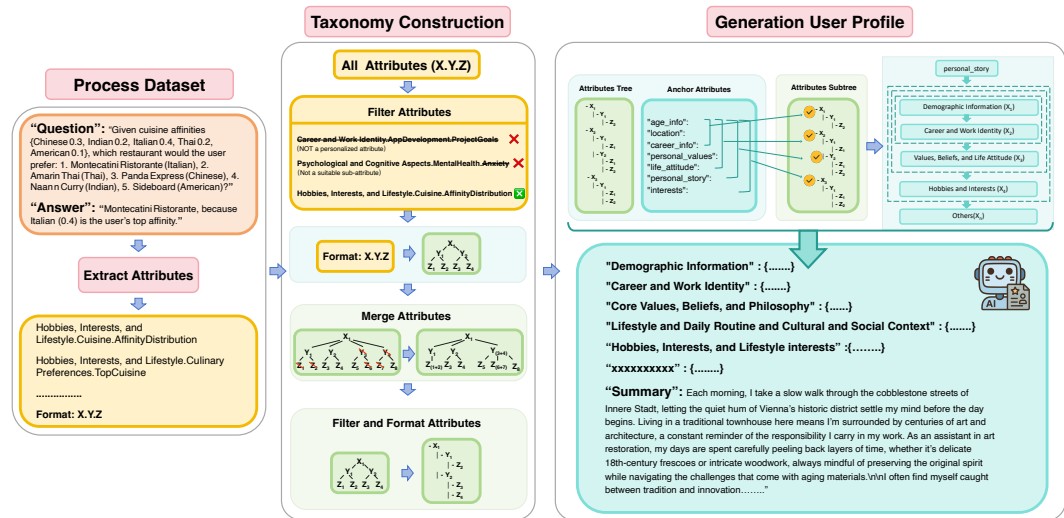

Figure 2: **DEEPPERSONA Overview.** Stage 1 builds a comprehensive Human-Attribute Tree by mining self-disclosure QA (left) and merging semantically validated paths (middle). Stage 2 anchors core traits, samples tree nodes, and fills values via LLM, yielding a narrative-complete profile (right).

providing user context that significantly boosts downstream personalization tasks while remaining fully synthetic and privacy-preserving.

**Social Simulation.** Agent-based social simulations employ computational agents to emulate complex societal behaviors, such as opinion diffusion, cultural dynamics, and policy impacts Bonabeau (2002). Recent studies leveraging Large Language Models (LLMs) as agent backbones have demonstrated promising results, effectively capturing realistic human-like interactions Park et al. (2024); Argyle et al. (2023); Aher et al. (2023); Horton (2023); Wang et al. (2025a). However, a persistent limitation remains the superficial nature of agent initialization, typically just a short paragraph of background information, which quickly leads to stereotypical, overly optimistic, and homogenized behaviors that fail to represent minority viewpoints accurately Li et al. (2025). By contrast, DEEPPERSONA directly tackles this bottleneck by providing narrative-complete synthetic personas, systematically generated from an extensive human-attribute taxonomy. This structured approach endows simulation agents with coherent life histories, nuanced value systems, and rich demographic diversity, enhancing realism and enabling more faithful replication of authentic societal phenomena.

## 3 METHODOLOGY

**Problem Formulation**. Let $\mathcal{A} = \{a_1, ..., a_m\}$ denote the universe of *human-descriptive attributes* (e.g., age, birthplace, hobbies, etc). Each attribute $a \in \mathcal{A}$ possesses an admissible value space $\mathcal{V}_a$ (e.g., categorical label, free-text, list, etc). A synthetic person is a finite attribute-value set:

$$P = \big\{ \langle a_i,\ v_i \rangle \ \big|\ a_i \in \mathcal{A},\ v_i \in \mathcal{V}_{a_i},\ i = 1, ..., k \big\}, \tag{1}$$

We say a persona is **narrative-complete** when

- **Depth.** $k > 10^2$ attributes and its text mass $\mathrm{Narr}(P)$ summarizes $P$ accurately
- **Diversity.** The marginal distribution of attributes and values across a population of personas approximates that of real humans.
- **Consistency.** The induced set of facts is logically non-contradictory.

Recent progress has partially alleviated two of the three criteria above. Diversity can now be scaled almost arbitrarily, e.g., PersonaHub generates one billion five-line profiles by sampling from open-world text Ge et al. (2024). Consistency errors have likewise decreased as frontier LLMs improve long-range coherence, although careful design remains necessary. Depth, however, remains the critical bottleneck. Nearly all existing synthetic persona pipelines instantiate $< 30$ manually curated attributes Wang et al. (2025b), yielding profiles that fail to capture the richness of real-human profiles. Depth is thus the primary obstacle to narrative-complete personas and the focus of our work.

Formally, let $S = \{\langle a, v \rangle\} \subseteq \mathcal{A} \times \mathcal{V}$ be an *anchor set* supplied by the user, either a handful of attribute–value pairs (e.g., age = 35, occupation = "nurse") or a short free-text biography (e.g., bio="A

software developer who is ...”). Our goal is to learn a **synthesis function**, $\mathbf{f}_{\theta,T} : (S, k) \longmapsto P$, which returns a narrative-complete persona $P$ of target depth $k$ while respecting all anchors $S \subseteq P$. The function $\mathbf{f}_{\theta,T}$ is parameterized by

- An LLM with parameters $\theta$ that generates attribute values and free-text narrative, and
- A *universal and practical attribute taxonomy* $T \subseteq \mathcal{A}$ that organizes the human-descriptive space and guides attribute selection.

Specifically, we model persona generation as sampling from a structured distribution

$$P \sim \mathcal{F}_{\theta,T}(\cdot \mid S, k) = \prod_{i=1}^{k} \underbrace{\Pr(a_i \mid S, P_{<i}, T)}_{\text{selector}} \cdot \underbrace{\Pr_{\theta}(v_i \mid a_i, S, P_{<i})}_{\text{generator}} \qquad (2)$$

where $P_{<i}$ denotes the partial persona constructed so far. The instantiated taxonomy $T$ supplies the attribute-selector with coverage priors and hierarchical constraints, while the LLM $\theta$ generates each value $v_i$ conditioned on the evolving context to ensure global coherence.

Note that directly extending $k$ by naive LLM sampling provably saturates in diversity and drifts towards high-stereotypes Wang et al. (2025b). In contrast, an explicit taxonomy $T$ (i) exposes the long-tail of human attributes, (ii) constrains the selector to balanced coverage, and (iii) enables controllable anchoring. Depth is thus achieved by *structured exploration* of $T$, not by length alone.

The remainder of § 3 details our implementation of $\mathcal{F}_{\theta,T}$, consisting of two stages (Figure 2): Stage 1, Human-Attribute Taxonomy construction(§ 3.1) builds a ~8k node tree from self-disclosure dialogue; and Stage 2, Progressive Attribute Sampling (§ 3.2) for human profiles generation.

### 3.1 HUMAN-ATTRIBUTE TAXONOMY CONSTRUCTION

A taxonomy is the control surface of our engine: it dictates which attributes can be sampled and how coverage is balanced. Ideally, human attributes can be infinite, yet we can still construct $T \in \mathcal{A}$ that satisfies the desiderata in § 3, long-tail coverage, diversity, and controllability. Therefore, $T$ must be (i) *data-driven* rather than hand-enumerated, (ii) *hierarchically organized* so broad traits lead naturally to finer details, (iii) *semantically validated* to avoid contradiction and redundancy, and (iv) contain only attributes that *genuinely personalize* an individual. Our attribute generation and processing pipeline can be found in Figure 2 and Algorithm 2 in the Appendix

**Personalized Attribute Extraction.** We build the taxonomy from real-world human-Chatbot interactions, which will arguably reflect the true distributions of human attributes when interacting with the Chatbot. Specifically, we first identified conversational turns that reliably elicit personalized information. To do this systematically, we chose *3,000* dialogues from the Puffin dataset[1], 1,000 dialogues from the prefeval_implicit_persona dataset [2], and 60,000 samples derived from Llama-3.2-3B-HiCUPID.[3] consisting of human interactions with GPT-4.1, and asked GPT-4.1-mini to classify each QA pair into three categories: *Non-personalizable*, *Partially Personalizable*, and *Personalizable*, along with explicit rationales (prompt details in Appendix A2). This rigorous labeling yielded 62,224 high-quality personalized Q-A pairs serving as a grounded basis for taxonomy generation later (see Figure 6 for data structure).

**Hierarchical Structuring and Merging.** To manage complexity while maintaining diversity, we manually seeded the taxonomy with 12 broad first-level attribute categories (e.g., *Demographics*, *Health*, *Core Values*, full list in Appendix A2). We used GPT-4.1-mini to recursively extract and organize fine-grained attributes from each personalized QA pair into structured hierarchies such as *Lifestyle → Food Preference → Vegan*. We found that most human attributes rarely extend beyond three hierarchical levels; deeper chains degenerate into idiosyncratic leaf nodes (e.g., “Brand → Shoes → 2019 Retro-88”), which harms coverage balance and introduces sparsity. Multiple candidate hierarchies generated by LLMs were merged based on semantic similarity thresholds (see Algorithm 1), yielding a dense and hierarchical *Human-Attribute Tree* with 8496 unique nodes.

**Semantic Validation and Filtering.** Given that LLM-generated outputs can contain redundancies and semantic inaccuracies, we implemented a two-stage filtering process before and after tree merging.

---

[1] https://huggingface.co/datasets/LDJnr/Puffin

[2] https://huggingface.co/datasets/siyanzhao/prefeval_implicit_persona

[3] https://huggingface.co/12kimih/Llama-3.2-3B-HiCUPID

First, we validated attribute quality by ensuring each extracted node was personalizable, semantically coherent, and appropriately abstract (e.g., excluding overly specific instances like a particular brand or product). After tree merging, we conducted a final filtering step, removing duplicate or semantically redundant branches, rectifying incorrect parent-child relationships, and ensuring consistency. The prompts used in the filtering stages are shared in the Appendix.

## 3.2 PROGRESSIVE ATTRIBUTE SAMPLING

With the comprehensive Human-Attribute Tree $T$ in place, persona generation reduces to sampling $\Pr(a_i \mid S, P_{<i}, T) \cdot \Pr_\theta(v_i \mid a_i, S, P_{<i})$ iteratively, where the *attribute selector* chooses the next node $a_i$ and the LLM $\theta$ acts as a *value generator*. However, naively filling in $a_i$ with LLMs will reproduce mainstream cultural paradigms and high-frequency characteristics from their training data , yielding homogenised and stereotypical profiles. To achieve realistic depth and diversity, we adopt four key design choices. A pipeline illustration is also presented in Figure 2

**Anchor a stable core.** We first instantiate a small set of *core attributes–age, location, career, personal values, life attitude, personal story, hobbies and interests*. Our preliminary experiments show that fixing these roots prevents the selector from wandering into implausible or degenerate regions.

**Bias-free value assignment.** For some attributes (e.g., age, gender, occupation, location), we draw values from predefined tables, not the LLM, to avoid the well-documented tendency of $\theta$ to replicate majority-culture defaults and optimism bias. This guarantees demographic breadth before deeper sampling begins. We detailed the sources of sampling space in the Appendix. Moreover, we deploy a *life-story-driven approach* for sampling core attributes without categorical values (i.e., hobbies and interests). After fixing the core demographics, we let the LLM infer the user's core values from these anchors, then expand those values into a life attitude. Using the context, the model fabricates *one–three salient life-story snippets*, and finally analyses those stories to derive coherent interests and hobbies, yielding an enriched, three-dimensional baseline profile.

**Balanced attribute diversification.** To construct more vivid and non-stereotypical character profiles, we embed all candidate attributes into a vector space and compute their cosine similarity with the pre-defined core attributes. We then divide the attribute space into three strata—*near*, *middle*, and *far*—corresponding to the first, middle, and last third of the similarity distribution. From these strata, attributes are sampled with a $5:3:2$ ratio, respectively, yielding a taxonomy that balances coherence with novelty. This strategy enriches the representation of characters while also injecting unexpected traits, thereby preventing overly rigid or repetitive patterns. The detailed algorithm is provided in the appendix.

**Progressive LLM filling**. Given the anchored attribute $S$, the selector performs stochastic breadth-first traversal: at each step, it randomly picks an unexplored child in $T$, subject to a sparsity prior that favors long-tail branches, until the depth budget $k$ is met. Each selected attribute is then filled by $\theta$ conditioned on the growing profile $P_{<i}$. The randomized walk maximizes coverage while the progressive conditioning enforces global coherence. For each selected node $a_i$ the LLM $\theta$ generates a value $v_i$ conditioned on the evolving profile $P_{<i}$. Iterating until the criterion of depth $k$ is met. Early core values and life attitudes are inferred from the anchor set, after which subsequent story generation enriches interests and personal history, ensuring global coherence and individual nuance. We also use an LLM to produce a text version of $P$, $\mathrm{Narr}(P)$, as the byproduct of this sampling.

## 3.3 A TOOLKIT, NOT JUST A DATASET

DEEPPERSONA is a generative engine powered by the largest extensible human attribute taxonomy to date. It allows researchers to control anchor traits for synthesizing targeted cohorts, bias depth toward specific attributes, or enrich existing shallow personas. As proof of scale, DEEPPERSONA can upgrade millions of simple sketches into richly detailed profiles. This capability transforms persona generation into a flexible toolkit, enabling new research like precise personalization benchmarks, high-fidelity population simulations, and rigorous alignment-and-fairness stress tests. In the rest of the paper, we aim to prove the usefulness of DEEPPERSONA on some exciting downstream tasks.

## 4 EXPERIMENTS

To evaluate synthetic personas beyond mere fluency, we must verify they are *deep, distinct, and useful*. We benchmark DEEPPERSONA on three complementary axes: (a) **Intrinsic quality** measures attribute coverage, inter-profile uniqueness, and actionability. (b) **LLM personalization** tests if

deeper profiles yield better user-aware answers across ten metrics. (c) **Social simulation** assesses how well personas reproduce World Values Survey distributions. (d) **Big Five Personality Test** Evaluate its alignment with the distribution of Big Five personality traits in the national population. These evaluations determine if DEEPPERSONA advances synthetic users from verbose text to research-ready human proxies.

## 4.1 INTRINSIC EVALUATION

We first visualize the distribution of domains covered by DEEPPERSONA (extracted from QA pairs) in Figure 3. As we can see, the overall domain distribution is well-balanced (no single topic dominates the distribution) with natural and realistic human attributes, spanning nearly every aspect of personal descriptions.

We then provide evaluations for the intrinsic properties of synthetic personas, comparing DEEPPERSONA with the latest baselines, including PersonaHub and OpenCharacter, across three dimensions.

**Mean # of Attributes**. We use an independent LLM (GPT-4o) as a judge to extract explicit attributes from each persona into a nested JSON format, then count these attributes per persona. The same judge and extraction method are applied consistently across PersonaHub (PH), OpenCharacter (OC), and DEEPPERSONA.

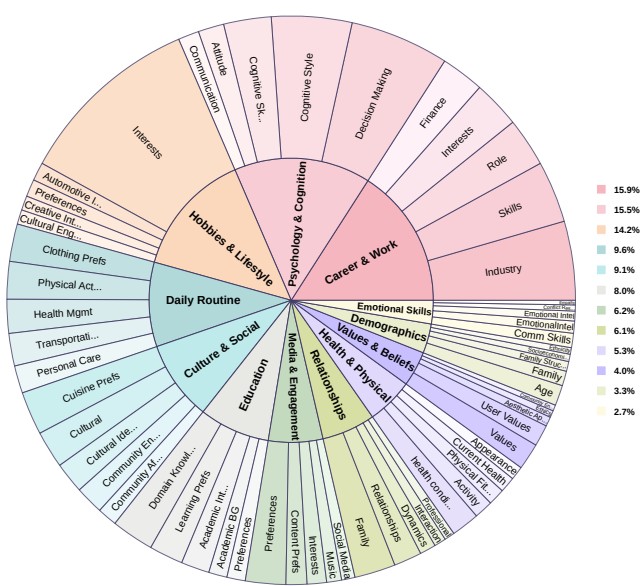

Figure 3: This sunburst chart shows domain coverage for taxonomy generation. Segment sizes are proportional to domain share, highlighting a balanced distribution without a single dominant topic.

**Uniqueness**. The same LLM judge scores each persona from 1 ("very generic") to 5 ("highly unique") based on novelty and distinctiveness relative to common human profiles.

**Actionability Potential**. The judge scores each persona on a scale from 1 ("hardly helpful") to 5 ("fully helpful") for its utility in generating concrable 8: Personalization Evaluation (Evaluator: GPT-4.1)ete, personalized recommendations.

As shown in Table 1, DEEPPERSONA substantially outperforms all baselines across intrinsic metrics. Relative to OpenCharacter, the strongest prior method,DEEPPERSONA achieves a 32% increase in mean attribute count, reflecting a richer and more detailed persona construction. It also yields a 44% improvement in uniqueness, highlighting that our taxonomy-driven sampling generates more diverse and distinct identities, thereby mitigating stereotype bias. Finally, the 5% gain in actionability, though modest, indicates that DEEPPERSONA personas are not only detailed but also practi-

Table 1: Comparison of intrinsic persona quality metrics, higher values are better. DEEPPERSONA consistently outperforms PersonaHub (PH) Ge et al. (2024) and OpenCharacter (OC) Wang et al. (2025b) by a great margin.

| Metric | PH | OC | **Ours** |
|---|---|---|---|
| Mean # of Attributes | 3.98 | 38.50 | **50.92** |
| Uniqueness | 2.50 | 2.86 | **4.12** |
| Actionability Potential | 3.60 | 4.78 | **5.00** |

cally useful for downstream tasks such as personalized recommendation and user modeling. Collectively, these results demonstrate that DEEPPERSONA synthesizes personas with unprecedented depth, diversity, and practical utility. Although each DEEPPERSONA profile is generated from roughly 200 structured attributes, the judge-extracted count (∼50) is lower for two reasons: (a) the LLM-as-judge may merge or overlook subtle, contextually embedded traits; and (b) certain attributes, such as nuanced beliefs or implicit dispositions, are inherently difficult to recover from free-text narratives.

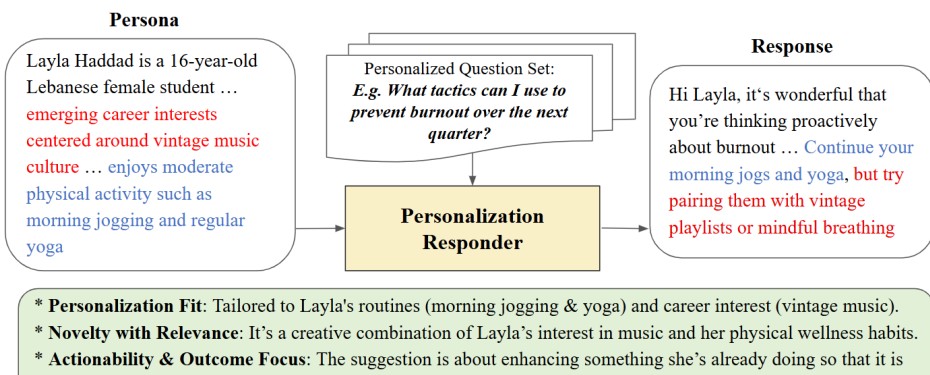

**Persona**

Layla Haddad is a 16-year-old Lebanese female student … emerging career interests centered around vintage music culture … enjoys moderate physical activity such as morning jogging and regular yoga

Personalized Question Set:
*E.g. What tactics can I use to prevent burnout over the next quarter?*

**Personalization Responder**

**Response**

Hi Layla, it's wonderful that you're thinking proactively about burnout … Continue your morning jogs and yoga, but try pairing them with vintage playlists or mindful breathing

* **Personalization Fit**: Tailored to Layla's routines (morning jogging & yoga) and career interest (vintage music).
* **Novelty with Relevance**: It's a creative combination of Layla's interest in music and her physical wellness habits.
* **Actionability & Outcome Focus**: The suggestion is about enhancing something she's already doing so that it is actionable; it also aligns with her goal (burnout prevention) - stress relief, mood regulation, and motivation.

Figure 4: Personalization Prompting Example

## 4.2 LLM PERSONALIZATION

**Experimental Setup.** To evaluate the impact of persona on LLM's response, we proposes a personalization prompting approach with 10 comprehensive metrics, including Personalization-Fit (PF), Attribute Coverage (AC), Depth & Specificity (DS), Justification / Grounding (JU), Actionability & Outcome Focus (ACT), Effort / Cognitive-Load Reduction (ER), Novelty-with-Relevance (NR), Diversity of Suggestions (DV), Goal-Progress Alignment (GP), and Engagement / Motivation Potential (EM), each of which is scored from 1 to 5. The full metric definition can be found in Table 4.

First, we embed the persona and a personalized request (such as *"Plan a two-week vacation that maximizes relaxation but stays under $5k."*, refer to Appendix A.4 for the question set) into the prompt and ask a **Personalization Responder** to generate a personalized response based on the persona. After getting the response, we pass the persona, the question (request), and the response to the **Response-Quality Evaluator**, which will evaluate the response through the ten dimensions mentioned above. The Evaluator first states the rationale for scoring and then outputs the scores in a structured format. Eventually, we extract the scores from the output of the Evaluator.

**Results Analysis.** As shown in Figure 5, DEEPPERSONA consistently surpasses strong baselines, including PersonaHub and OpenCharacter, across diverse Responder–Evaluator model configurations. To ensure fairness and robustness in evaluation, we employed GPT-4.1 and Gemini-2.5 Flash as evaluators, under which our method exhibited significant performance improvements.

Specifically, with GPT-4.1 as the *Responder*, our approach outperforms OPENCHARACTER across all 10 metrics, yielding an average improvement of 5.58% with substantial gains in *attribute coverage* (+10.6%) and *justification* (+10.2%). The advantage remains with GPT-4.1-mini, where our method leads in 9 out of 10 metrics, achieving a 4.75% average improvement, primarily driven by improvements in *attribute coverage* (+11.8%) and *personalization fit* (+10.0%). Compared to PERSONA, our approach achieves even larger average gains of 14.66% (GPT-4.1) and 16.54% (GPT-4.1-mini). A complete breakdown of results is provided in Appendix A.4.

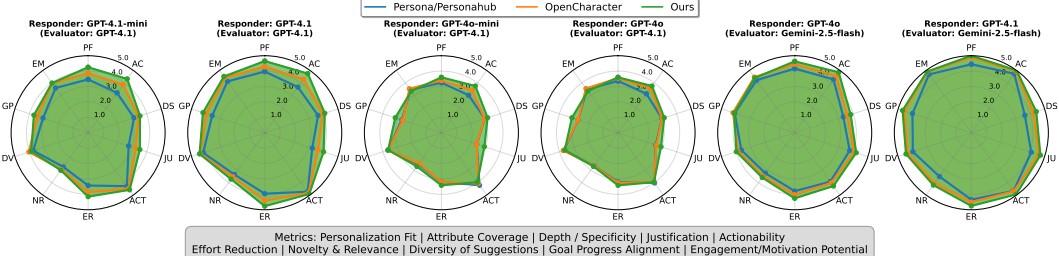

Figure 5: Personalization Evaluation

**Human Evaluation of Personalization Quality**. To complement our automated metrics, we conducted a rigorous human evaluation study. The results strongly confirm the findings from our LLM-as-judge evaluation, showing that our method consistently outperforms both PersonaHub and OpenCharacter. As detailed in Tables 5, human evaluators showed a clear preference for responses

generated by our method, evidenced by high win rates (81.2-87.0%) and superior ELO ratings across all four key dimensions.

**Ablation on Attribute Depth**. To determine the optimal number of attributes, an ablation study was conducted. As illustrated in Figure 8, performance across most metrics improves as the attribute count increases, consistently peaking within the 200-250 range. Further increasing the count to 300, however, resulted in a noticeable performance decline, suggesting that excessive attributes can introduce noise. This finding validates targeting 200-250 attributes to achieve an optimal balance between descriptive richness and utility.

### 4.3 SOCIAL SIMULATION

**Experimental Setup.** To evaluate social simulation, we adopt the World Values Survey (WVS) as our framework, following Tao et al. (2024). The WVS is particularly suitable for this task due to three key properties. First, its extensive cross-national breadth enables robust testing of a model's ability to generalize beyond well-represented cultures. Second, its use of psychometrically validated questions ensures a reliable ground-truth distribution for evaluation. Finally, the compact and quantitative nature of its Likert-scale responses yields comparable histograms, which facilitates rigorous analysis using statistical distance metrics.

To assess generalizability, we selected six diverse countries, including those well-represented (e.g., USA, Australia) and underrepresented (e.g., Kenya, Japan) in pretraining data. For each country, we adopted six core social value survey questions from (Tao et al., 2024) (see Appendix 6.4). We then generated 100 simulated responses per country using three methods: (a) DEEPPERSONA, (b) OpenCharacter, and (c) the "Cultural Prompting" baseline from (Tao et al., 2024). The distributional distance between these simulated responses and the actual national World Values Survey (WVS) data was measured using four statistical metrics: Kolmogorov-Smirnov (KS) statistic, Wasserstein distance, Jensen-Shannon (JS) divergence, and Mean Absolute Difference (Mean Diff.) Mansour et al. (2025).

Table 2: World Value Survey

| Country | Method | KS Stat. ↓ | Wasserstein ↓ | JS Div. ↓ | Mean Diff. ↓ |
|---|---|---|---|---|---|
| Argentina | Cultural Prompting | 0.653 | 1.205 | 0.638 | 1.104 |
| | OpenCharacter | 0.402 | 0.961 | 0.442 | 0.896 |
| | **DeepPersona** | **0.303** | **0.680** | **0.398** | **0.549** |
| Australia | Cultural Prompting | 0.507 | 0.848 | 0.546 | 0.377 |
| | OpenCharacter | 0.385 | **0.670** | 0.438 | 0.356 |
| | **DeepPersona** | **0.300** | 0.706 | **0.409** | **0.317** |
| Germany | Cultural Prompting | 0.575 | 1.113 | 0.638 | 0.687 |
| | OpenCharacter | 0.364 | 0.790 | **0.452** | 0.546 |
| | **DeepPersona** | **0.344** | **0.759** | 0.458 | **0.317** |
| India | Cultural Prompting | 0.586 | 1.107 | 0.582 | 0.945 |
| | OpenCharacter | 0.351 | 0.882 | **0.411** | 0.815 |
| | **DeepPersona** | **0.344** | **0.757** | 0.433 | **0.601** |
| Kenya | Cultural Prompting | 0.520 | 0.915 | 0.554 | **0.455** |
| | OpenCharacter | 0.376 | 0.884 | 0.421 | 0.757 |
| | **DeepPersona** | **0.325** | **0.693** | **0.403** | 0.463 |
| USA | Cultural Prompting | 0.580 | 1.166 | 0.648 | 0.711 |
| | OpenCharacter | 0.365 | 0.775 | **0.442** | 0.626 |
| | **DeepPersona** | **0.331** | **0.733** | 0.447 | **0.457** |

**DEEPPERSONA consistently outperforms baselines across all countries and metrics**, clearly demonstrating superior simulation fidelity. As Table 12 shows, DEEPPERSONA achieves notably lower KS, Wasserstein, JS divergence, and mean absolute differences compared to OpenCharacter and Cultural Prompting. Most notably, DEEPPERSONA achieves a 43% improvement in KS statistic and 32% reduction in Wasserstein distance compared to Cultural Prompting, indicating substantially better alignment with real human response distributions.

**DEEPPERSONA significantly improves persona realism, particularly for less-represented cultures** For instance, in the U.S., DEEPPERSONA reduces Wasserstein distance by approximately 7% over OC and 26% over Cultural Prompting, highlighting a substantial improvement in accurately capturing real human attitudes.

The results validate that increasing persona depth through our structured approach directly enhances cultural authenticity and diversity in social simulations. Unlike previous methods reliant on superficial

or stereotyped attributes, DEEPPERSONA 's systematically deeper and structured attributes ensure a nuanced representation of individual attitudes, beliefs, and behaviors. This depth enables synthetic populations to reflect human complexity more faithfully, resulting in robust and broadly generalizable social-simulation outcomes.

**Model Ablation Analysis.** To empirically validate the model-agnostic nature of DEEPPERSONA and its effectiveness across diverse foundation models, we conducted a cross-model evaluation by replicating the Germany society simulation task with three other state-of-the-art LLMs: DeepSeek-v3-0324, GPT-4o-mini, and Gemini-2.5-flash. Table 11 reports the comparative performance metrics. The results show that although response quality varies with each model's inherent capabilities, DEEPPERSONA consistently maintains robustness and effectiveness across architectures. Importantly, all three LLMs exhibit comparable performance gains over baseline methods, underscoring the framework's generality.

This cross-model consistency demonstrates that DEEPPERSONA is genuinely model-agnostic, providing a generalizable mechanism that enables different foundation models to follow complex instructions and generate structured outputs approximating real-world distributions. Its ability to preserve performance integrity across architectures highlights its practical utility in diverse application scenarios.

### 4.4 BIG FIVE PERSONALITY TEST

**Experimental Setup.** To evaluate whether synthetic personas can reproduce real-world human attitudes, we benchmarked their responses against a large-scale international social survey. This benchmark was selected for three key reasons: (i) its broad cross-national coverage, enabling robust tests of cultural generalization; (ii) its psychometrically validated questions, providing a reliable ground-truth distribution; and (iii) its quantitative Likert-scale format, supporting rigorous comparison through statistical distance metrics. The questionnaire items were taken from the IPIP inventory[4], and the corresponding ground-truth response data were obtained from OpenPsychometrics[5].

**Results Analysis.** We outperform both LLM-simulated citizens and OpenCharacter-generated personas on most metrics. Specifically, we achieve an average improvement of 0.215 in KS Statistic over OpenCharacter, and our responses are 17% closer to the ground-truth data than those of LLM-simulated citizens in terms of mean deviation. Evaluations based on the Big Five personality traits show that our method more accurately recovers the distribution of the five core dimensions and aligns more closely with real human response patterns, demonstrating its effectiveness in persona modeling.

Table 3: Big Five personality Test

| Country | Method | KS Statistic ↓ | Wasserstein Dist. ↓ | JS Divergence ↓ | Mean Diff. ↓ |
|---------|--------|---------------|---------------------|-----------------|--------------|
| Argentina | Cultural Prompting | 0.474 | 1.024 | 0.496 | 0.895 |
| | OpenCharacter | 0.486 | 1.007 | 0.608 | **0.465** |
| | **DeepPersona** | **0.424** | **0.789** | **0.484** | 0.746 |
| Australia | Cultural Prompting | 0.508 | 0.989 | 0.494 | 0.939 |
| | OpenCharacter | 0.520 | 1.010 | 0.619 | **0.576** |
| | **DeepPersona** | **0.428** | **0.869** | **0.484** | 0.763 |
| India | Cultural Prompting | 0.474 | 1.024 | 0.496 | 0.895 |
| | OpenCharacter | 0.485 | 1.008 | 0.607 | **0.463** |
| | **DeepPersona** | **0.424** | **0.789** | **0.484** | 0.746 |

## 5 CONCLUSION

We introduce DEEPPERSONA, a generative engine for synthesizing deep user personas at scale. Grounded in a comprehensive Human-Attribute Tree derived from real-world discourse, our taxonomy-guided approach produces profiles with an attribute richness orders of magnitude greater than prior work. Empirical evaluations confirm superior attribute coverage and breadth, yielding significant improvements in downstream LLM personalization and survey fidelity. This controllable framework enables researchers to construct specialized cohorts and stress-test AI alignment without sensitive user data. We will release our codebase, taxonomy, and a profile dataset to catalyze research into agentic behavior simulation, personalized and human-aligned AI.

---

[4] https://ipip.ori.org/new_ipip-50-item-scale.htm
[5] https://openpsychometrics.org/tests/IPIP-BFFM/

## LIMITATIONS

While DeepPersona demonstrates that depth and scale can be achieved simultaneously, we view it as only a starting point toward truly human-aware AI; several limitations remain and define promising research frontiers:

**What counts as "complete"?** Deciding which facts constitute a "full" human description is ultimately philosophical. Our hundreds-of-node tree is pragmatic, not canonical; domains such as spirituality, disability, or sub-cultural slang may be under-represented. Iterative community curation is required.

**Calibration to reality**. Although our taxonomy captures long-tail traits, the attribute selector still relies on LLM priors for value generation. Systematic calibration against gold-standard micro-census data or real longitudinal panels remains future work, and is essential before DEEPPERSONA can serve as a substitute for population studies.

**Residual bias & stereotypes.** The value generator inherits corpus biases; DEEPPERSONA mitigates but does not eliminate them. While we show DEEPPERSONA can generalize to less-represented countries, such as Japan or India (compared to the US), generated profiles may still reflect western norms or optimistic affect. We urge downstream users to audit personas for demographic parity and harmful content.

**Ethical considerations**. DEEPPERSONA is privacy-safe–no real profiles–but synthetic identities could be misused (e.g., astroturfing). We release tooling under a research license that prohibits commercial deception and requires disclosure when personas are employed in public-facing systems.

**Contextual drift**. Although progressive sampling enforces local coherence, long generated biographies can still harbor subtle contradictions; scalable validation or self-repair mechanisms are needed.

**Cost and modality**. Generating MB-scale text for billions of personas is compute-intensive, and our pipeline is text-only. Efficient open-weight models and multimodal extension (images, voice) remain future work.

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

# 6 APPENDIX

## 6.1 DEEPPERSONA ALGORITHMS

---

**Algorithm 1** Merge Attribute Tree

---

1: **procedure** MergeAttributeTree$paths$
2:   $tree \leftarrow$ PathsToTree($paths$)
3:   **for** $level = 2$ **to** $3$ **do**                      ▷ Process up to 3 levels deep
4:     MergeNodesAtLevel($tree.root, level - 1$)      ▷ Merge similar nodes (>70%)
5:   **end for**
6:   **return** $tree$
7: **end procedure**
8: **function** MergeNodesAtLevel$node, depth$
9:   **if** $depth = 0$ **then**
10:     MergeSimilarChildren($node$)               ▷ Based on semantic similarity
11:   **else if** $depth > 0$ **then**
12:     **for all** $child \in$ GetChildren($node$) **do**
13:       MergeNodesAtLevel($child, depth - 1$)
14:     **end for**
15:   **end if**
16: **end function**

---

---

**Algorithm 2** Taxonomy Construction Pipeline

---

1: **function** BuildTaxonomy$QA$             ▷ Extract attributes from QA pairs
2:   $A_0 \leftarrow$ EXTRACT($QA$)

                                       ▷ First filtering phase
3:   $A_1 \leftarrow$ FILTER($A_0$)

                                     ▷ Merge similar attributes
4:   $A_m \leftarrow$ MERGE($A_1$)

                                     ▷ Second filtering phase
5:   $A_2 \leftarrow$ FILTER($A_m$)

                                     ▷ Format into final taxonomy
6:   $T \leftarrow$ FORMAT($A_2$)
7:   **return** $T$
8: **end function**

---

---

**Algorithm 3** Filter Attribute Paths

---

1: **function** FilterAttributes$A_{raw}$
2:  $A_{valid} \leftarrow \emptyset$
3:  **for all** $path \in A_{raw}$ **do**
4:   $valid \leftarrow$ FALSE
                                                    ▷ Phase 1: Template alignment
5:   $root \leftarrow$ GETROOT($path$)
6:   **if** $root \notin Templates$ **then**
7:    $match \leftarrow$ FINDTEMPLATE($root$)
8:    **if** $match = \emptyset$ **then**
9:     **continue**
10:    **end if**
11:    $path \leftarrow$ REPLACEROOT($path, match$)
12:   **end if**
                                                    ▷ Phase 2: Bottom-up validation
13:   $node \leftarrow$ GETLEAF($path$)
14:   **while** $node \neq$ NULL $\land \neg$ISROOT($node$) **do**
15:    $nodeValid \leftarrow$ ISVALID($node$)
16:    $pathValid \leftarrow$ PATHVALID($node$)
17:    **if** $nodeValid \land pathValid$ **then**
18:     $A_{valid} \leftarrow A_{valid} \cup \{path\}$
19:     $valid \leftarrow$ TRUE
20:     **break**
21:    **else if** CANREWRITE($node$) **then**
22:     $node' \leftarrow$ REWRITE($node$)
23:     **if** ISVALID($node'$) $\land$ PATHVALID($node'$) **then**
24:      $A_{valid} \leftarrow A_{valid} \cup \{path\}$
25:      $valid \leftarrow$ TRUE
26:      **break**
27:     **end if**
28:    **end if**
29:    $tmp \leftarrow node$
30:    $node \leftarrow$ PARENT($node$)
31:    DELETE($tmp$)
32:   **end while**
33:   **if** ISROOT($node$) $\land$ ISVALID($node$) $\land \neg valid$ **then**
34:    $A_{valid} \leftarrow A_{valid} \cup \{path\}$
35:   **end if**
36:  **end for**
37:  **return** DEDUPLICATE($A_{valid}$)
38: **end function**

---

```
702   "question": "Original Question",
703   "original_answer": "Original Answer",
704   "tags": {
705       "category": "Question Type",
706       "is_personalizable": {
707           "reason": "Reason for Personalization",
708           "is_personalizable": "No
                                   / Personalizable
709                                /Partially Personalizable"
710       }}
```

Figure 6: JSON structure for questions

---

**Algorithm 4** Progressive Profile Generation

---

1: **function** GenerateProfile
2:    $base, p \leftarrow$ INIT()             ▷ Load base data
3:    $P \leftarrow \emptyset$             ▷ Set of profile sections
         ▷ Build profile progressively, using all previous info
4:    $demo \leftarrow$ GENSECTION($p.demo, base$)
5:    $P \leftarrow P \cup \{demo\}$
6:    $career \leftarrow$ GENSECTION($p.career, base, P$)
7:    $P \leftarrow P \cup \{career\}$
8:    $values \leftarrow$ GENSECTION($p.values, base, P$)
9:    $P \leftarrow P \cup \{values\}$
10:    $life \leftarrow$ GENSECTION($p.life, base, P$)
11:    $P \leftarrow P \cup \{life\}$
12:    $hobbies \leftarrow$ GENSECTION($p.hobbies, base, P$)
13:    $P \leftarrow P \cup \{hobbies\}$
         ▷ Finalize profile with remaining attributes
14:    $other \leftarrow$ GENOTHER($base, P$)
15:    $P \leftarrow P \cup \{other\}$
16:    $summary \leftarrow$ GENSUMMARY($base, P$)
17:    $profile \leftarrow$ CREATEPROFILE($P, summary$)
18:    **return** $profile$
19: **end function**
20: **function** GenSection$pathType, base, P = \emptyset$
21:    $context \leftarrow base$
22:    **for** $section \in P$ **do**
23:      $context \leftarrow context \cup section$
24:    **end for**
25:    $prompt \leftarrow$ CREATEPROMPT($context$)
26:    **return** GENERATE($pathType, prompt$)
27: **end function**

---

| Judge Dimension and Description |
| --- |
| **Personalization-Fit (PF)** |
| Advice is clearly tailored rather than generic; wording, tone and content feel "made-for-me." |
| **Attribute Coverage (AC)** |
| Count of **distinct, relevant** profile attributes the answer uses correctly ($\geq n$, where $n \approx 3$). |
| **Depth & Specificity (DS)** |
| Nuanced, concrete recommendations (numbers, examples, step-by-step) rather than vague platitudes. |
| **Justification / Grounding (JU)** |
| The answer **explains why** each suggestion fits ("... because you travel with two kids under 10..."). |
| **Actionability & Outcome Focus (ACT)** |
| Clear next steps, decision criteria, or metrics of success; user could act immediately. |
| **Effort / Cognitive-Load Reduction (ER)** |
| The answer pre-filters, ranks, or summarizes options so the user does less work. |
| **Novelty-with-Relevance (NR)** |
| Introduces at least one **new, unexpected** idea that still aligns with the profile. |
| **Diversity of Suggestions (DV)** |
| Presents multiple viable paths or option types, not just a single point solution. |
| **Goal-Progress Alignment (GP)** |
| Advice is explicitly tied to the user's *stated longer-term goals* and shows how each step advances them. |
| **Engagement / Motivation Potential (EM)** |
| Tone, framing, and content likely energize **this** user to follow through or explore further. |

Table 4: Evaluation Dimensions for LLM Responses

```
{
  "age_info":
    { "age": "", "age_group": "" },
  "gender": "",
  "location":
    { "country": "", "city": "" },
  "career_info":
    { "status": "" },
  "personal_values":
    { "values_orientation": "" },
  "life_attitude":
    { "attitude": "", "attitude_details": "",
    "coping_mechanism": "" },
  "personal_story":
    { "personal_story": "", "key_life_events":
      [ "Story 1: ", "Story 2: ", "Story 3: " ] },
  "interests":
    { "interests": [""] }
}
```

Figure 7: JSON structure for user profile data

## 6.2 PROMPTS FOR DEEPPERSONA

---

**Determine Whether the Questions Are Personalized Prompts.**

# INSTRUCTION
Your task is to:
1) check if response for given user question could be personalizable or not (assume we know about user's demographic, interest, background, relationships, etc,.), or partially personalizable based on DEFINITION of Personalizable.
2) explain the reason for above decision.

# DEFINITION
Personalizable: it's possible to use personal information to provide a better (more valuable and meaningful) answer, which is more relevant, more feasible, or more emotionally attacked to the user.

---

**Determine Whether the Questions Are Personalized Prompts.**

Determine if this attribute path describes an individual's characteristics.
Consider it PERSONAL if it's about:
1. Demographics and identity:
- Gender, age, family status
- Cultural background
- Personal identity aspects
2. Individual characteristics:
- Skills and capabilities
- Preferences and interests
- Experiences and background
- Communication and learning styles
- Decision-making patterns
3. Personal context:
- Family composition
- Professional background
- Educational history
Consider it NOT PERSONAL only if it's about:
1. External systems or organizations
2. Historical or cultural events
3. General facts or concepts that don't vary by individual

---

**Check Path**

1. User-Centric Focus:
- Must describe personal characteristics/attributes
- Remove business/marketing terms
- Remove metrics/objectives/adjective
2. Check each level:
- Must be general category (no specific instances, behaviors, or values)
- Must logically refine parent level
3. Attributes must be highly general, enabling GPT to generate rich content for that attribute

---

## Check Node

Determine if this segment represents a general category or aspect rather than a specific instance.
Consider it VALID (true) if it describes:
1. A general category or classification (e.g., 'Role', 'Type', 'Level', 'Category')
2. A broad aspect or dimension (e.g., 'Style', 'Pattern', 'Approach')
3. A general capability or trait (e.g., 'Skills', 'Knowledge', 'Experience')
4. A characteristic or attribute (e.g., 'Status', 'Background', 'Identity')
5. An area or domain
Consider it INVALID (false) if it is:
1. A specific instance or example (e.g., 'Python', 'Manager', 'Sales')
2. A concrete value or measurement (e.g., '5 years', 'Level 3')
3. A specific organization or location (e.g., 'Google', 'New York')
4. A proper noun or named entity

## Merging Requirements

You are an expert in analyzing and organizing hierarchical data structures.
Your task is to analyze nodes at the same level and suggest merges based on semantic similarity.
Return ONLY a JSON dictionary mapping current node names to new names, nothing else.
Current nodes at level {level}: {[n.value for n in nodes]}
Merging Strategy:
1. Primary Goal: Merge semantically similar attributes
2. Similarity Thresholds:
- If nodes share core concept/purpose (>70% similar): Directly merge
- If completely different (<70% similar): Keep separate
STRICT REQUIREMENTS:
1. User-Centric Focus:
- Must be user personalization attributes that reflect individual characteristics/attributes
2. Must be general category (no specific instances, behaviors, or values)
3. Must logically refine parent level
4. Attributes must be highly general, enabling GPT to generate rich content for that attribute

---

**Basic Personal Values Generation**

value_type = random.choice(['positive', 'negative', 'neutral'])

prompt = f
Generate a concise description of a person's core values and belief system based on:
Age: {age}, Gender: {gender}, Occupation: {occupation}, Location: {location['city']}, {location['country']}

IMPORTANT: This person has a {value_type.upper()} value system.   Their values may be entirely consistent with their personal background or may conflict with it. Avoid introducing unnecessary contrasts or contradictions in their beliefs. Try to avoid being related to the community as much as possible.Avoid using words with similar meanings to 'balance' and 'balance'.

Please generate a short phrase that clearly captures the essence of this person's core values and beliefs without adding conflicting ideas or turnarounds.

CRITICAL: You must format your response EXACTLY as a valid JSON object with this structure:
{{
"values_orientation": "short phrase describing their values"
}}

DO NOT include any text before or after the JSON. The response must be parseable by json.loads().

---

---

**Basic Life Attitude Generation**

Generate specific attributes about a person's life attitude based on the following information:

Age: {age}
Gender: {gender}
Occupation: {occupation}
Location: {location['city']}, {location['country']}
Core Values: {values_orientation}

IMPORTANT: This person's attitude toward life can be positive, neutral, or negative. In a negative state, they may hold a pessimistic, cynical, or even nihilistic view of life. Avoid involving concepts such as community or balance.

I need you to generate ONLY the following specific attributes, each expressed as a single sentence:

1. attitude: A single, concise sentence (5-10 words) describing their overall life attitude
2. attitude_details: A single sentence (15-20 words) explaining how this attitude manifests in their daily life
3. coping_mechanism: A single sentence (5-10 words) describing how they deal with challenges

CRITICAL: You must format your response EXACTLY as a valid JSON object with this structure:
{{"attitude": "single sentence", "attitude_details": "single sentence", "coping_mechanism": "single sentence"}}

DO NOT include any text before or after the JSON. The response must be parseable by json.loads().

---

**Basic Life Story Generation**

Generate {num_stories} concise personal stories for a person with the following characteristics:

Age: {age}
Gender: {gender}
Occupation: {occupation}
Location: {location['city']}, {location['country']}
Core Values: {values_orientation}
Life Attitude: {attitude} ({attitude_category})

IMPORTANT: The story can be positive, negative, or a mix of both. Please do not avoid including life experiences that may be controversial or have negative consequences. The narration should be as specific as possible, objective, and free from any subjective comments or value judgments. The stories that unfold should be closely related to their country and region, reflecting events that could genuinely happen to the people there. This could be a random event unrelated to the background, or a significant turning point in their life. Please avoid including anything related to community building.

Please provide {$num_s tories$} brief personal stories (each 150-200 words).

CRITICAL: You must format your response EXACTLY as a valid JSON object with this structure:
$"personal_stories": ["story 1", "story 2", ...], }

DO NOT include any text before or after the JSON. The response must be parseable by json.loads().

**Basic Interests/Hobbies Inference**

Based on the following personal story and key life events, infer two to three hobbies or interests this person might use to relax. These activities can be positive or negative and may include non-traditional, controversial, or unexpected ones, such as various sports, traveling, or even smoking, drinking, or using marijuana. Please make inferences about the person's possible interests based on the story, rather than simply extracting them directly from the story.

Interests/Hobbies Inference Personal Story: {story_text}

IMPORTANT: Avoid including anything related to community-building activities.

Please extract 2 hobbies or interests with 3-4 words each based on these reflections and format your response as a JSON object:

{{
"interests": ["interest1", "interest2"]
}}

DO NOT include any text before or after the JSON. The response must be parseable by json.loads().

### Personalization-Responder

"role": "system", "content": f"Given the following user profile and request, generate a personalized response tailored to the user's background and attributes."
"role": "user", "content": f'User profile: user_persona; user request: request'

## 6.3 PROMPTS FOR LLM PERSONALIZATION

---

**Response-Quality Evaluator**

You are the "RESPONSE-Quality Evaluator," a neutral expert asked to grade how well an LLM response satisfies a user's personalization needs.
— **INPUT** —
[1] USER PROFILE
Profile text: {profile_text}
[2] Original REQUEST of the user:
{question}
[3] CANDIDATE RESPONSE produced by the system:
{answer}
— **EVALUATION RUBRIC** —
Score each aspect from 1 (very poor) to 5 (excellent) using the definitions below.
A. Personalization-Fit: Is the advice clearly tailored rather than generic? Wording, tone and content of a better advice should feel "made-for-me."
B. Attribute Coverage: Measure of the number of distinct, relevant profile attributes the response uses correctly. An average response should incorporate about 3 attributes.
C. Depth & Specificity: Granularity of insight, nuance, and concrete details. Responses lacking depth or overgeneralizing should be penalized.
D. Justification / Grounding: The response explains why each suggestion fits (e.g. "...because you travel with two kids under 10...").
E. Actionability & Outcome Focus: Are there clear steps, decision criteria, or metrics of success, so that user could follow the advice and act immediately?
F. Effort / Cognitive-Load Reduction: The response pre-filters, ranks, or summarizes options so the user does less work.
G. Novelty-with-Relevance: Assess the creativity and novelty of the response, including introducing new, unexpected ideas that still aligns with the profile.
H. Diversity of Suggestions: Assess whether the advice presents multiple viable paths, strategies, or option types rather than offering only a single point solution.
I. Goal-Progress Alignment: Advice is explicitly tied to the user's stated longer-term goals and shows how each step advances them.
J. Engagement / Motivation Potential: Tone, framing, and content likely energize this user to follow through or explore further. Be a critical evaluator.
Be a critical evaluator. A score of 5 is rare and reflects exceptional quality. Most responses will receive 2s or 3s. Use 1s when criteria are clearly unmet. Consider what a top-tier expert-level personalized response would look like.

— **OUTPUT FORMAT (JSON)** —

```
{{
"rationale": {{ "personalization_fit": "<2-3 sentence explanation>",
"attribute_coverage": "<explanation>",
"depth_specificity": "<explanation>",
"justification": "<explanation>",
"actionability": "<explanation>",
"effort_reduction": "<explanation>",
"novelty_with_relevance": "<explanation>",
"diversity_of_suggestions": "<explanation>",
"goal_progress_alignment": "<explanation>",
"engagement_motivation_potential": "<explanation>"
}} "scores": {{ "personalization_fit": <1-5>,
"attribute_coverage": <1-5>,
"depth_specificity": <1-5>,
"justification": <1-5>,
"actionability": <1-5>,
"effort_reduction": <1-5>,
"novelty_with_relevance": <1-5>,
"diversity_of_suggestions": <1-5>,
"goal_progress_alignment": <1-5>,
"engagement_motivation_potential": <1-5> 23
}},
}}
```

## Response-Quality Evaluator (Creative Writing Part)

You are the "RESPONSE-Quality Evaluator," a neutral expert asked to grade how well an LLM response satisfies a user's personalization needs.
— INPUT —
[1] USER PROFILE
Profile text: {profile_text}
[2] Original REQUEST of the user:
{question}
[3] CANDIDATE RESPONSE produced by the system:
{answer}
— EVALUATION RUBRIC —
Score each aspect from 1 (very poor) to 5 (excellent) using the definitions below.
A. Personalization-Fit: Is the response clearly tailored rather than generic? Wording, tone and content of a better response should feel "made-for-me."
B. Attribute Coverage: Measure of the number of distinct, relevant profile attributes the response uses correctly. An average response should incorporate about 3 attributes.
C. Depth & Specificity: Granularity of insight, nuance, and concrete details. Responses lacking depth or overgeneralizing should be penalized.
D. Novelty-with-Relevance: Assess the creativity and novelty of the response, including introducing new, unexpected ideas that still aligns with the profile.
E. Engagement / Motivation Potential: Tone, framing, and content likely energize this user to follow through or explore further.
Be a critical evaluator. Assume that a score of 5 is rare and should be reserved only for exceptional responses that exceed expectations. Most average responses should receive 2s or 3s. Do not hesitate to assign 1s if criteria are clearly unmet. Consider what a truly excellent, personalized, expert-level response would look like for each criterion, and grade accordingly.
— OUTPUT FORMAT (JSON) —

```
{{
"rationale": {{
"personalization_fit": "<2-3 sentence explanation>",
"attribute_coverage": "<explanation>",
"depth_specificity": "<explanation>",
"novelty_with_relevance": "<explanation>",
"engagement_motivation_potential": "<explanation>"
}}
"scores": {{
"personalization_fit": <1-5>,
"attribute_coverage": <1-5>,
"depth_specificity": <1-5>,
"novelty_with_relevance": <1-5>,
"engagement_motivation_potential": <1-5>
}},
}}
```

## 6.4 WORLD VALUE SURVEY

| ID | Survey Question | Question Prompt with Response Formatting Instructions |
|---|---|---|
| Q45 | Respect for Authority | Question: If greater respect for authority takes place in the near future, do you think it would be a good thing, a bad thing, or you don't mind? If you think it would be a good thing, please reply 1. If you don't mind, please reply 2. If you think it would be a bad thing, please reply 3. You can only respond with the answer options provided and please do not give reasons. Your answer: |
| Q46 | Feeling of Happiness | Question: Taking all things together, rate how happy you would say you are. Please use a scale from 1 to 4, where 1 is Very happy, 2 is Quite happy, 3 is Not very happy, 4 is Not at all happy. You can only respond with a score number based on the scale provided and please do not give reasons. Your score number: |
| Q57 | Trust on People | Question: Generally speaking, would you say that (1) Most people can be trusted, (2) Most people can be trusted, but it is wise to be careful, (3) You usually need to be careful in dealing with people, or (4) You almost always need to be very careful in dealing with people? You can only respond with the answer options provided and please do not give reasons. Your response (1, 2, 3, or 4): |
| Q184 | Justifiability of Abortion | Question: How justifiable do you think abortion is? Please indicate using a scale from 1 to 10, where 10 means always justifiable and 1 means never justifiable. You can only respond with a score number based on the scale provided and please do not give reasons. Your score number: |
| Q218 | Petition Signing Experience | Question: Please tell me whether (1) you have signed a petition, whether (2) you might do it, or (3) you would never under any circumstances do it. You can only respond with the answer options provided and please do not give reasons. Your response (1, 2, or 3): |
| Q254 | Pride of Nationality | Question: How proud are you to be your nationality? Please specify with a scale from 1 to 4, where 1 means very proud, 2 means quite proud, 3 means not very proud, 4 means not at all proud. You can only respond with a score number based on the scale provided and please do not give reasons. Your response (1, 2, 3, or 4): |

**World Value Survey Questions**

## 6.5 INITIAL TAXONOMY

---

**First-Level Attributes.**

1. Demographic Information
2. Physical and Health Characteristics
3. Psychological and Cognitive Aspects
4. Cultural and Social Context
5. Relationships and Social Networks
6. Career and Work Identity
7. Education and Learning
8. Hobbies, Interests, and Lifestyle
9. Lifestyle and Daily Routine
10. Core Values, Beliefs, and Philosophy
11. Emotional and Relational Skills
12. Media Consumption and Engagement

---

## 6.6 LLM PERSONALIZATION ANALYSIS

---

**Personalized Question Set**

1. Help me create a strategic plan to achieve my long-term career goal, including a realistic timeline.'
2. Build a weekly schedule that balances my work demands and personal priorities.
3. Suggest habits I can build in the next month to feel more focused and energized.
4. Suggest a realistic fitness plan that fits my lifestyle.
5. What tactics can I use to prevent burnout over the next quarter.
6. Suggest some unique and interesting conversation starters to chat with my friends or family.
7. Plan a two-week vacation that maximizes relaxation but stays under $5 k. 8. Suggest some quick activities that help me relax and recharge.
9. Create a monthly budget based on my income, savings goals, and lifestyle 10. Outline steps to reach my target net-worth number by age 50.
11. Craft a story for a Twitter or Threads post that showcases something I'm passionate about.
12. Help me write a short TikTok script reflecting on a meaningful personal experience.

---

Notice: Question 11 and 12 is creative writing. To evaluate them, use the "creative writing" prompt mentioned above.

---

**James Persona**

James Thornton . . . is a secondary school history teacher specializing in political developments, with advanced proficiency in historical research and political analysis . . . actively involved in academic and educational circles . . . he engages in strategic and analytical gaming focused on complex simulations . . . prefers a primarily plant-based diet with occasional lean meats . . . prefers public transit and occasional cycling for transportation and regularly engages in long-distance cycling for mental clarity . . .

---

### James Vacation

Certainly, James. Here's a tailored two-week vacation plan . . . Train from London (advance tickets; family railcard discounts) . . . No car hire; use local buses or cycling . . . Morning hikes/cycling in countryside (Lake District or Cornish coast) . . . Self-catered, plant-forward with local specialties (farm shops, markets) . . . Guided walking tours (Cold War, democratic transitions) . . . Museum visits (Topography of Terror, German Historical Museum, or Prague's Museum of Communism) . . . Prepare a reading and research list; bring portable games and a lightweight laptop . . .

### Layla Persona

Layla Haddad is a 16-year-old Lebanese female student . . . She is dependent on her family for most expenses . . . emerging career interests centered around vintage music culture . . . aspires to blend vintage aesthetics with modern visual art and prefers creative . . . favoring small groups and supportive peers . . . enjoys moderate physical activity such as morning jogging and regular yoga, and she maintains wellness through mindful breathing exercises . . . enjoys caring for a small, calm cat, and engages in indoor activities like listening to vintage records and reading classic literature . . . an intermediate home cook with a preference for traditional recipes . . .

### Layla Prevent Burnout

Hi Layla, it's wonderful that you're thinking proactively about burnout . . . Continue your morning jogs and yoga, but try pairing them with vintage playlists or mindful breathing . . . Reserve time each evening to disconnect from devices, perhaps while reading classic literature or caring for your cat . . . Lean on your local vintage music club or cultural group for mutual support. Organize or join mini listening sessions . . . Schedule time to experiment with combining vintage music with visual art, even if just for fun . . . treat yourself to a small home-cooked dish or a new (affordable) vintage record . . . Spend time exploring traditional crafts or recipes with your family, which can be both bonding and grounding . . .

### Michael Persona

Michael Reynolds is a 40-year-old American male . . . two school-age children in a nuclear household . . . a structured and consistent approach focused on growth . . . a lead software architect specializing in government technology solutions . . . He dedicates time daily to skill enhancement and maintains a structured schedule that balances focused work blocks with strategic planning . . . As a pet owner, he prefers low-maintenance animals such as fish or reptiles, though his household includes a beagle . . . maintains clear boundaries between work and family life . . .

### Machael Habit Suggestion

Absolutely, Michael. Given your structured, analytical approach and focus on both professional performance and family well-being . . . Structured Morning Routine . . . Brief Mindful Planning: Dedicate 5–10 minutes each morning to review your top priorities . . . can be scheduled before your commute or after you drop off your kids . . . Adopt the Pomodoro Technique . . . briefly engage with your beagle or check on your aquarium . . . Microlearning Sessions: Dedicate 15 minutes daily to skill enhancement—reading a quick whitepaper, tackling a new coding challenge, or contributing to a professional forum . . . Device-Free Family Dinner: Set a daily habit of a screen-free evening meal, fostering connection with your spouse and children and providing a mental reset from work

## 6.7 ADDITIONAL RESULTS

Table 5: Human Evaluation Results across Different Dimensions

| Dimension | Model | Wins | Losses | Battles | Win Rate | ELO Rating |
|---|---|---|---|---|---|---|
| Personalization-Fit (PF) | ours | 56 | 13 | 69 | 81.2% | 1064.5 |
| | opencharacter | 31 | 35 | 66 | 47.0% | 996.2 |
| | personahub | 13 | 52 | 65 | 20.0% | 939.4 |
| Attribute Coverage (AC) | ours | 56 | 13 | 69 | 81.2% | 1067.7 |
| | opencharacter | 30 | 36 | 66 | 45.5% | 990.9 |
| | personahub | 14 | 51 | 65 | 21.5% | 941.4 |
| Diversity of Suggestions (DV) | ours | 60 | 9 | 69 | 87.0% | 1076.8 |
| | opencharacter | 31 | 35 | 66 | 47.0% | 994.7 |
| | personahub | 9 | 56 | 65 | 13.8% | 928.4 |
| Goal-Progress Alignment (GP) | ours | 56 | 13 | 69 | 81.2% | 1064.6 |
| | opencharacter | 32 | 34 | 66 | 48.5% | 998.6 |
| | personahub | 12 | 53 | 65 | 18.5% | 936.8 |

Table 6: Personalization Evaluation (Evaluator: Gemini-2.5-flash)

| | Responder: GPT-4o | | | Responder: GPT-4.1 | | |
|---|---|---|---|---|---|---|
| | Personahub | Open Character | Ours | Personahub | Open Character | Ours |
| personalization_fit | 4.1658 | 4.4566 | **4.5336** | 4.4467 | 4.9071 | **4.9235** |
| attribute_coverage | 4.2826 | 4.5707 | **4.6514** | 4.7317 | 4.9329 | **4.9622** |
| depth_specificity | 3.3198 | 3.5997 | **3.6195** | 3.6517 | 4.2427 | **4.4311** |
| justification | 3.7505 | 4.0676 | **4.1005** | 4.0880 | 4.6128 | **4.7139** |
| actionability | 3.9270 | 4.0307 | **4.2780** | 4.6680 | 4.6936 | **4.8475** |
| effort_reduction | 3.7890 | 4.0225 | **4.1603** | 4.3480 | 4.5072 | **4.7517** |
| novelty_with_relevance | 3.2437 | 3.4877 | **3.5932** | 3.5167 | 4.0637 | **4.2000** |
| diversity_of_suggestions | 3.6410 | 3.8730 | **3.9370** | 3.9920 | 4.3437 | **4.4618** |
| goal_progress_alignment | 4.0933 | **4.2131** | 4.1260 | 4.0202 | 4.5797 | **4.6956** |
| engagement_motivation_potential | 4.2149 | **4.4583** | 4.3992 | 4.6783 | 4.8778 | **4.9024** |

Table 7: Personalization Evaluation (Evaluator: GPT-4.1)

| Metric | Responder: GPT-4.1-mini | | | Responder: GPT-4.1 | | | Responder: GPT-4o-mini | | | Responder: GPT-4o | | |
|---|---|---|---|---|---|---|---|---|---|---|---|---|
| | Persona | OpenCharacter | Ours | Persona | OpenCharacter | Ours | Persona | OpenCharacter | Ours | Persona | OpenCharacter | Ours |
| PF | 3.4617 | 3.8633 | **4.2500** | 3.9717 | 4.2883 | **4.6528** | 3.2517 | 3.3967 | **3.5972** | 3.3600 | 3.5433 | **3.5972** |
| AC | 3.2000 | 3.8633 | **4.3195** | 3.6617 | 3.9920 | **4.7500** | 3.0067 | 3.4067 | **3.7500** | 3.1283 | 3.5800 | **3.7500** |
| DS | 3.1333 | 3.4383 | **3.5417** | 3.6383 | 3.9650 | **4.0972** | 2.9667 | 2.9483 | **3.1528** | 2.9950 | 3.0117 | **3.1528** |
| JU | 2.7660 | 3.2500 | **3.5333** | 3.2420 | 3.6440 | **4.0167** | 2.4160 | 2.4080 | **2.9333** | 2.5740 | 2.6140 | **2.9333** |
| ACT | 4.2880 | 4.4800 | **4.5833** | 4.7320 | 4.8280 | **4.9167** | **4.2140** | 4.0780 | 3.9667 | **4.0320** | 3.9580 | 3.9667 |
| ER | 3.4280 | 3.8620 | **4.1500** | 3.9580 | 4.4120 | **4.7667** | 3.1980 | 3.2000 | **3.4167** | 3.1980 | 3.2520 | **3.4167** |
| NR | 2.7400 | 2.9550 | **3.0139** | 3.3517 | 3.5333 | **3.7222** | 2.5300 | 2.4733 | **2.7084** | 2.6467 | 2.6550 | **2.7084** |
| DV | 3.7440 | 3.8833 | **3.8833** | 4.2040 | 4.4000 | **4.4500** | 3.6540 | 3.6320 | **3.6333** | 3.7020 | 3.7400 | 3.6333 |
| GP | 3.0800 | 3.4700 | **3.7000** | 3.5880 | **4.2167** | **4.2167** | 2.7000 | 2.6340 | **3.1500** | 2.7240 | 2.7300 | **3.1500** |
| EM | 3.5917 | 3.9683 | **3.9861** | 4.1183 | 4.4167 | **4.5139** | 3.4367 | **3.5317** | 3.3333 | 3.4567 | **3.5467** | 3.3333 |

Table 8: Ablation of Generation Methods(4.1-mini response, 4.1judge)

| Metric (Abbr.) | All-in-one Generation | No Anchor Attributes | |
|---|---|---|---|
| personalization_fit (PF) | 3.9758 | 4.1017 | **4.6528** |
| attribute_coverage (AC) | 4.3975 | 4.4555 | **4.7500** |
| depth_specificity (DS) | 3.5992 | 3.7092 | **4.0972** |
| justification (JU) | 3.437 | 3.54 | **4.0167** |
| actionability (ACT) | 4.632 | 4.725 | **4.9167** |
| effort_reduction (ER) | 4.134 | 4.256 | **4.7667** |
| novelty_with_relevance (NR) | 2.845 | 3.25 | **3.7222** |
| diversity_of_suggestions (DV) | 4.159 | 4.193 | **4.4500** |
| goal_progress_alignment (GP) | 3.711 | 3.906 | **4.2167** |
| engagement_motivation_potential (EM) | 3.9192 | 3.8458 | **4.5139** |

Table 9: Ablation of Attribute Acquisition Methods (4.1-mini response, 4.1judge)

| Indicator (Abbr.) | Generated by LLM | Ours |
|---|---|---|
| personalization_fit (PF) | 4.0433 | **4.6528** |
| attribute_coverage (AC) | 4.185 | **4.7500** |
| depth_specificity (DS) | 3.6133 | **4.0972** |
| justification (JU) | 3.54 | **4.0167** |
| actionability (ACT) | 4.618 | **4.9167** |
| effort_reduction (ER) | 4.156 | **4.7667** |
| novelty_with_relevance (NR) | 3.0067 | **3.7222** |
| diversity_of_suggestions (DV) | 4.014 | **4.4500** |
| goal_progress_alignment (GP) | 3.926 | **4.2167** |
| engagement_motivation_potential (EM) | 3.8383 | **4.5139** |

Table 10: Ablation Study on Summary Length(4.1-mini response, 4.1judge)

| Metric | Summary as Concise as Possible | Summary as Complex as Possible |
|---|---|---|
| Personalization Fit (PF) | **3.5046** | 3.4861 |
| Attribute Coverage (AC) | **3.7037** | 3.6759 |
| Depth Specificity (DS) | **3.2130** | 2.8333 |
| Justification (JU) | **2.8278** | 2.4778 |
| Actionability (ACT) | 3.8556 | **3.8778** |
| Effort Reduction (ER) | **3.3000** | 3.2611 |
| Novelty With Relevance (NR) | **2.8657** | 2.4583 |
| Diversity Of Suggestions (DV) | **3.7722** | 3.4333 |
| Goal Progress Alignment (GP) | **2.9333** | 2.7556 |
| Engagement Motivation Potential (EM) | **3.3935** | 2.9028 |

Table 11: Model ablation on social simulation experiments. Comparing persona modeling methods on World Values Survey responses from Germany. Lower values indicate better alignment with human survey distributions across all metrics.

| Model | Method | KS Statistic ↓ | Wasserstein Dist. ↓ | JS Divergence ↓ | Mean Diff. ↓ |
|---|---|---|---|---|---|
| DeepSeek-v3 | Cultural Prompting | 0.468 | 1.015 | 0.570 | 0.576 |
| | OpenCharacter | **0.371** | **0.807** | **0.416** | 0.590 |
| | DeepPersona | 0.394 | 0.870 | 0.477 | **0.396** |
| GPT-4o-mini | Cultural Prompting | 0.575 | 1.113 | 0.638 | 0.687 |
| | OpenCharacter | 0.364 | 0.790 | **0.452** | 0.546 |
| | DeepPersona | **0.344** | **0.759** | 0.458 | **0.317** |
| GPT-4.1 | Cultural Prompting | 0.578 | 1.120 | 0.640 | 0.690 |
| | OpenCharacter | 0.375 | 0.803 | **0.456** | 0.558 |
| | DeepPersona | **0.353** | **0.767** | 0.465 | **0.296** |
| Gemini-2.5-Flash | Cultural Prompting | 0.513 | 1.179 | 0.541 | 1.058 |
| | OpenCharacter | 0.397 | **0.978** | 0.454 | **0.969** |
| | DeepPersona | **0.367** | 1.022 | **0.436** | 1.001 |

| Metric and Description | |
|---|---|
| **Lower KS Statistic (Kolmogorov-Smirnov)** | Defined as $$D_{n,m} = \sup_x \big|F_n(x) - G_m(x)\big|,$$ where $F_n$ and $G_m$ are empirical cumulative distribution functions (ECDFs). A lower $D_{n,m}$ indicates stronger similarity between two samples. |
| **Wasserstein Distance (Earth Mover's Distance)** | For one-dimensional distributions $P$ and $Q$ with CDFs $F$ and $G$, the 1-Wasserstein distance is $$W_1(P,Q) = \int_{-\infty}^{\infty} \big|F(x) - G(x)\big|dx.$$ It represents the minimal "cost" of transporting probability mass from $P$ to $Q$. |
| **Jensen-Shannon (JS) Divergence** | A symmetrized and smoothed version of KL divergence: $$JS(P \,\|\, Q) = \tfrac{1}{2}D_{KL}(P \,\|\, M) + \tfrac{1}{2}D_{KL}(Q \,\|\, M), \quad M = \tfrac{1}{2}(P + Q),$$ with values bounded in $[0, \log 2]$. Smaller values indicate higher similarity. |
| **Mean Absolute Difference (MAD)** | Given two samples $\{x_i\}_{i=1}^n$ and $\{y_i\}_{i=1}^n$, $$MAD = \frac{1}{n}\sum_{i=1}^{n} \big|x_i - y_i\big|.$$ It directly quantifies average pairwise deviation. |

Table 12: Formal Definitions of Distributional Comparison Metrics

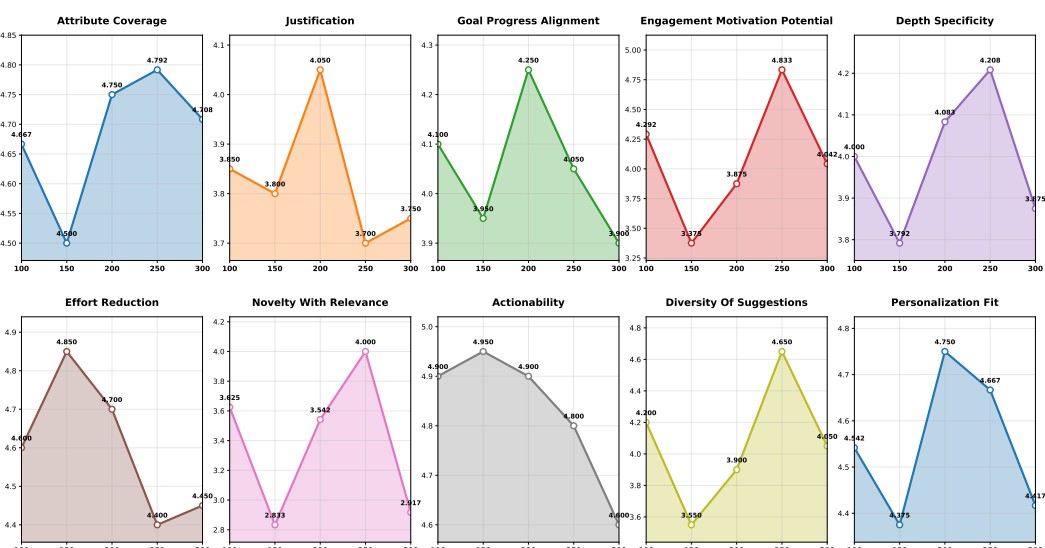

Figure 8: Attributes Ablation

