# OpenReview forum: "DeepPersona: A Generative Engine for Scaling Deep Synthetic Personas"
_ICLR.cc/2026/Conference — Submitted to ICLR 2026_

### Official Review · Reviewer_rXfh · 2025-10-28

**Soundness:** 4
**Presentation:** 4
**Contribution:** 4
**Rating:** 8
**Confidence:** 5

**Summary:**

This work introduces DeepPersona, a two-stage generative engine that synthesizes detailed, diverse, and customizable synthetic persona data. The authors first construct the largest human-attribute taxonomy to date by mining and filtering self-disclosure content from human-LLM interactions. They then employ a progressive attribute sampling approach that iteratively selects diverse attributes and conditions a large language model to generate coherent values and narrative text.

**Strengths:**

- The motivation is clear and important, pinpointing the problem of "persona depth" in previous persona generation approaches.
- The method the authors use to extract is systematic and thoughtful.
- Evaluation is done extensively in a multi-faceted manner, ranging from four different downstream tasks.
- Experiments are conducted on many frontier AI models from different sources, further supporting the generality of this work.
- Human experiments are included to complement the possible concerns regarding the instability of LLM judges.
- Most importantly, this work provides a scalable platform for synthetic persona generation, which I think is a significant contribution to the community.

**Weaknesses:**

I did not spot any significant weaknesses in this paper. One minor regret would be that qualitative examples are limited. It would be great to see qualitative comparisons between previous approaches and DeepPersona.

Also, this is minor, but there are some formatting issues on page 23. Please amend the overflow issue.

**Questions:**

N/A

---

> ### Author Response · Authors · 2025-11-25
> **Response to Reviewer rXfh: Thank You and Clarifications on Qualitative Examples**
>
> We sincerely thank Reviewer rXfh for the extremely positive assessment and strong support. We greatly appreciate your recognition of our contributions, including the systematic taxonomy construction, multi-faceted evaluations across frontier models, and the scalability of our generative engine. Your affirmation of our focus on “persona depth” strongly encourages us to further pursue this direction. Following your suggestion, we have added qualitative comparisons to better demonstrate DeepPersona’s advantages and have corrected the formatting issue accordingly.
>
> We also sincerely appreciate the reviewer’s valuable feedback and suggestions.
> The example shown here provides a simple comparison. Our version displays only the summary since the number of attributes is too large for the dialogue box. Detailed attribute values are available in our anonymous GitHub repository.
>
> ***Deepersona:*** A wholesaler with a wide range of products and the expertise to recommend the best options for the business,
>
> ***Opencharacter:*** Marcus Whitfield is a 42-year-old Caucasian male born in Chicago, Illinois. He stands 6 feet tall with a sturdy build that reflects his active lifestyle. He has short, neatly trimmed brown hair with a few strands of gray and piercing blue eyes that convey both warmth and sharp intelligence. His style of dress is business casual, usually consisting of well-fitted blazers, crisp shirts, and polished shoes, giving him a professional yet approachable appearance.
> Marcus has over 20 years of experience in the wholesale industry. He began his career in a small, family-owned wholesale business, where he learned the fundamentals from the ground up. Over time, he expanded his expertise across a wide range of product categories, from electronics to gourmet foods. His strong understanding of market trends and customer needs has made him a trusted advisor to businesses seeking to optimize their inventory strategies and product selections.
> In terms of personality, Marcus is known for his analytical mindset and his ability to build lasting relationships with clients. He is approachable, supportive, and generous with his knowledge, which makes him well-liked by both new and experienced business owners. His commitment to his profession is balanced by his personal life, where he enjoys mentoring young entrepreneurs, spending time with his family, and reading extensively to stay ahead in the constantly evolving wholesale market.
>
> ***Ours***  "In southern Albania, I’ve spent the past two decades living in Sarandë, though my roots are firmly planted in Fier. My days rarely follow a predictable rhythm—sometimes I’m at my desk before sunrise, reviewing regulatory updates or drafting compliance reports; other times, I’m on-site late into the night, overseeing technical inspections or troubleshooting a stubborn safety system. The hum of the plant, the sharp scent of metal and ozone, and the constant pressure of deadlines have become as familiar to me as the quiet of my own apartment.
> I grew up in a working-class family shaped by the aftermath of communism, where education was non-negotiable and emotional restraint was the norm. That upbringing left me with a taste for simple, home-cooked meals—beans, grilled vegetables, a little bread—and a habit of saving rather than spending. I keep my home sparse and functional, with a dedicated workspace and a reliable air filter, and I avoid clutter, pets, and unnecessary guests. After a long shift, I’ll often take a solitary walk through the quieter streets, letting the night air clear my head before settling in with a technical manual or a case study on power plant decommissioning.
>
> Mentoring younger engineers is one of the few social interactions I seek out, though I’m known for being blunt and sometimes impatient when it comes to sloppy work or shortcuts. I prefer direct communication, whether in Albanian or technical English, and I have little patience for empty rhetoric or superficial trends. My routines are strict—coffee for long nights, cigarettes when the stress spikes, and classical music or old Albanian songs in the background as I read.
> I rarely travel unless it’s for a site visit or a technical conference. Most of my leisure time is spent with logic puzzles, simulation games, or documentaries about crisis management. I recycle diligently, avoid waste, and keep my appliances energy-efficient—not out of fashion, but because I see it as my responsibility."
>
> To help reviewers better understand our profiles and their differences from prior work, we have prepared a GitHub repository with persona samples from PersonaHub, OpenCharacter, and DeepPersona.
> We hope these examples provide a clearer qualitative comparison and illustrate the characteristics of our approach.
> We sincerely appreciate the reviewer’s time and invite them to explore the examples here:
> https://anonymous.4open.science/r/iclr_differences-2DB1/README.md

---

### Official Review · Reviewer_8nFh · 2025-11-01

**Soundness:** 2
**Presentation:** 3
**Contribution:** 3
**Rating:** 4
**Confidence:** 4

**Summary:**

The paper introduces a taxonomy-guided framework that generates synthetic personas by sampling from a human-attribute tree. It addresses the deep and coherent persona generation from existing work. Experiments show higher personalization quality and closer alignment to human distributions on World Values Survey and Big Five benchmarks.

**Strengths:**

- The synthetic persona generation problem is interesting and timely.
- They provide a large-scale taxonomy of human attributes, which is really beneficial for the literature.
- The experiments are good, showing the advantages of the generated synthetic persona.
- The authors provide a method to diversify selected attributes.

**Weaknesses:**

- The method is naive. They did break the sampling procedure into two stages (sampling attributes from the taxonomy first and then sampling values from the given attributes), but they are heavily manually engineered.
- Generally, it seems to be a neat paper and can bring benefits to the community, but the novelty is limited. It would be more appreciated if this paper were submitted to the benchmark and dataset tracks instead of the main tracks.

**Questions:**

- Can we have a learnable way to learn the selector and generator so that it generates personas towards a chosen population? like these methods: PICLe: Eliciting Diverse Behaviors from Large Language Models with Persona In-Context Learning, ICML24 and Mixture-of-Personas Language Models for Population Simulation, ACL25.
- Why do we set the ratio to 5 : 3 : 2 ratio? Will any other combination work?

---

> ### Author Response · Authors · 2025-11-25
> **Response to Reviewer 8nFh‘s Weaknesses: Clarifying Methodological Design and Contribution Type**
>
> 1. We sincerely thank the reviewer for the thoughtful and constructive feedback, and for recognizing the potential value of our work to the broader research community. We truly appreciate the time and effort invested in carefully examining our manuscript and providing detailed comments. We would like to respectfully clarify that, although our approach is intentionally designed to be concise, structured, and interpretable, it is by no means “manually engineered” or trivial in nature. On the contrary, it is built upon a fully automated, scalable, and algorithmically grounded framework, aiming to ensure both scientific rigor and practical usability, as detailed below.
>
> Our method is not merely a dataset, but a generative engine. Compared with existing systems such as PersonaHub and OpenCharacter (which primarily serve as static datasets), our generation process is more principled and algorithmically grounded, encompassing taxonomy extraction and construction, hierarchical priors, and progressive generation. Specifically, Stage 1 of our framework can be used independently for attribute extraction and cleaning, enabling users to preprocess and refine attribute sets or taxonomy from various datasets. Stage 2, in turn, allows users to flexibly invoke any individual step to accomplish specific generation functions, or to combine both stages for end-to-end user-profile synthesis. This modular design provides both fine-grained control and full automation within the same generative engine, offering a flexible yet systematic way to generate coherent and diverse persona representations. Even in its most general configuration, our generated user profiles achieves superior performance across multiple evaluation metrics compared with PersonaHub and OpenCharacter.
>
> In addition, our two-stage design—covering both taxonomy construction and user profile generation—is fully automated, encompassing over 8,000 attribute nodes and capable of generating an arbitrary number of user profiles, all without any manual annotation or adjustment throughout the entire process.

---

> ### Author Response · Authors · 2025-11-25
> **Response to 8nFh‘s Questions: On Learnable Population-Guided Generation and Sampling Strategy**
>
> 2. **Can we have a learnable way to learn the selector and generator so that it generates personas towards a chosen population? like these methods: PICLe: Eliciting Diverse Behaviors from Large Language Models with Persona In-Context Learning, ICML24 and Mixture-of-Personas Language Models for Population Simulation, ACL25.**- We sincerely thank the reviewer for highlighting these relevant works. Both studies are excellent and highly valuable contributions to the field. While our current work cannot cover all related directions, we will carefully consider incorporating these approaches in our future research. Even though we did not perform model training in this work, our framework is still able to achieve strong and competitive results.
>
> Regarding PICLe (ICML’24), its goal is to more stably and accurately elicit target personas and behavioral styles from large language models. While PICLe focuses on behavioral steering, our work centers on constructing complex persona structures. Although the two works differ in focus, PICLe’s mechanism can complement our framework. Specifically, PICLe could serve as an upstream module in the base_info stage, generating initial drafts based on demographic factors. Our DeepPersona pipeline would then perform structural expansion and semantic deepening on these drafts to ensure both consistency and psychological depth in the resulting personas. Therefore, while PICLe can assist in the generation phase, it does not replace our core contribution in achieving multi-dimensional representation and long-term consistency.
>
> Regarding Mixture of Personas (MoP, ACL’25), its main contribution lies in guiding large language models to simulate human-like behaviors more realistically and diversely. However, its core objective fundamentally differs from ours. MoP focuses on group-level simulation to address the lack of diversity in standard LLM outputs, primarily using personas as a means to broaden response patterns. In contrast, our work tackles the limitation that existing methods are often too shallow to capture the complexity of real human characteristics. Our goal is not merely to increase quantity or diversity but to enhance fidelity, depth, and granularity. Unlike MoP’s “low-depth but high-quantity” strategy, our framework achieves both high depth and high diversity, generating profiles with rich and internally consistent structures. This explains why our approach performs strongly in downstream tasks—our profiles behave more like real humans, rather than simply producing diverse but superficial behavioral fragments.
>
>
> 3. **Why do we set the ratio to 5 : 3 : 2 ratio? Will any other combination work? - We have explored various other combinations, as shown in the table below.**
>
> We appreciate the reviewer’s question and agree that the choice of the 5:3:2 ratio warrants clarification. To validate this configuration, we conducted an ablation study varying the sampling ratios of near:middle:far attributes while keeping all other experimental settings fixed. Specifically, we used GPT-4.1-mini as the generation model and GPT-4.1 for evaluation, testing ratios of 7:2:1, 6:3:1, 4:3:3, and 3:3:4, in addition to the 5:3:2 configuration used in our paper.
>
> The results are summarized in (reproduced below). Across multiple evaluation dimensions — the 5:3:2 ratio achieves consistently strong and balanced performance, even if not the single best on every individual metric. In contrast, more extreme ratios (e.g., 7:2:1 or 3:3:4) tend to sacrifice either coverage or personalization quality. Therefore, we selected the 5:3:2 ratio as it provides the most stable overall trade-off between depth, diversity, and relevance across metrics.
>
> ### Table 6. Evaluation Metrics Across Different Ratios
>
> | Metric | 7:2:1 | 6:3:1 | 4:3:3 | 3:3:4 | Paper |
> |---------|------------|------------|------------|------------|-------|
> | Actionability | 4.4000 | 4.5125 | 4.6063 | 4.6875 | 4.5833 |
> | Attribute Coverage | 4.1823 | 4.3021 | 4.2188 | 4.3021 | 4.3195 |
> | Depth Specificity | 3.4115 | 3.5521 | 3.5781 | 3.4745 | 3.5417 |
> | Diversity of Suggestions | 4.0000 | 4.0500 | 3.9313 | 3.9156 | 3.8833 |
> | Effort Reduction | 3.8250 | 3.8875 | 4.0500 | 4.0594 | 4.1500 |
> | Engagement Motivation Potential | 3.9219 | 3.7292 | 3.7083 | 3.7578 | 3.9861 |
> | Goal Progress Alignment | 3.6500 | 3.9000 | 3.7750 | 3.7875 | 3.7000 |
> | Justification | 3.4562 | 3.5500 | 3.4938 | 3.5206 | 3.5333 |
> | Novelty with Relevance | 2.9688 | 2.8229 | 2.9115 | 2.9469 | 3.0139 |
> | Personalization Fit | 3.9844 | 4.0312 | 4.0260 | 4.1146 | 4.2500 |

---

> ### Author Response · Authors · 2025-11-29
> **Follow-Up Comment to Reviewer 8nFh: On Novelty, Core Contribution, and the Positioning of MoP**
>
> As the discussion phase nears its end, we wanted to briefly follow up to ensure that our previous responses have adequately addressed your concerns and express our gratitude for your constructive feedback.
>
> **Regarding the comparsiong with "Mixture-of-Personas Language Models for Population Simulation"**
>
> Before elaborating, we want to thank you for directing our attention to "Mixture-of-Personas Language Models for Population Simulation" (ACL '25). After a thorough reading, we find the framing of mixture-based population simulation is elegant, and MoP has set a high bar in formalizing how LLM behavior can be diversified and statistically grounded at a population level. **We will definitely include a formal citation and a detailed discussion of MoP in our final revision to properly acknowledge its contribution to the field.**
>
> However, precisely because MoP is such a strong and well-structured piece of work, the contrast with our method becomes even sharper--and highlights why the novelty concerns raised in the initial review may not fully reflect the actual nature of our contribution. In particular,
>
> - **MoP aims at group-level behavioral diversity**, leveraging mixtures of shallow personas to broaden a model’s output patterns across a population.
> - **Our method targets individual-level persona depth**, constructing multi-dimensional, internally coherent, psychologically grounded user profiles that remain consistent across long horizons.
> - **MoP operates primarily at the modeling level** (guiding LLM behavior through mixtures).
> - **Our system operates primarily at the representation level** (algorithmically generating the persona structures themselves through an automated taxonomy-driven engine).
>
> We genuinely admire MoP for what it set out to accomplish, and we view our approach as complementary: MoP focuses on _how models behave_, whereas we focus on _how personas are formed, structured, and grounded_. This distinction is fundamental to understanding the novelty of our work.
>
> -----
>
> **Regarding the novelty judgement**
>
> We respectfully believe that the characterization of our method as “naive” or primarily “manually engineered” may **understate both its algorithmic content and its contribution to the field (we also clarified this with great detail in the global response)**. As detailed in our previous comment, our framework is a fully automated generative engine with:
> - **A large-scale, automatically constructed taxonomy** of >8,000 attribute nodes, equipped with hierarchical priors.
> - A two-stage, end-to-end pipeline that **systematically connects taxonomy construction and persona synthesis**, not just as a static dataset, but as a configurable engine that can generate arbitrarily many coherent, high-depth profiles.
> - A modular design that **allows users to invoke individual components** (e.g., attribute extraction, cleaning, hierarchical sampling, or deep profile generation) or run the entire pipeline, providing both interpretability and flexibility rather than ad-hoc rules.
>
> In this sense, our contribution is **methodological and algorithmic** rather than merely curating or packaging existing data. While we fully agree that our approach is designed to remain interpretable and relatively simple to implement, this "simplicity" is the result of deliberate design choices and intellectual elegance (e.g., hierarchical sampling, progressive deepening) rather than manual, one-off engineering.
>
> Finally, because your summary and rating place substantial weight on “limited novelty,” we would be very grateful if you could let us know whether the above clarifications change your understanding of our main contribution. Any further clarification of your perspective on these points would be highly appreciated.
>
> Thanks.

---

### Official Review · Reviewer_ERkS · 2025-11-01

**Soundness:** 1
**Presentation:** 1
**Contribution:** 1
**Rating:** 2
**Confidence:** 3

**Summary:**

This paper introduces DEEPPERSONA, a two-stage generative framework for creating synthetic personas.

**Strengths:**

I cannot find any strength or contribution in the current manuscript of the paper.

**Weaknesses:**

The writing is misleading and difficult to follow, suggesting that the paper is not yet in a finished state.

In the abstract, the authors claim to be “mining thousands of real user–ChatGPT conversations.” However, Section 3 shows that no new data were mined; instead, the work relies entirely on existing datasets (Puffin, prefeval_implicit_persona). This inconsistency significantly weakens the claimed novelty.

At the start of Section 3, several important concepts are introduced without any explanation. Terms such as “text mass Narr(P)” and “persona/attribute depth” appear multiple times but are never defined or justified. Similarly, the choice of parameters—such as enforcing k > 10²—is arbitrary and unsupported by analysis or intuition. The authors should explain what k represents, why that threshold was chosen, and how it affects outcomes.

The paper also claims to contribute a dataset or toolkit (Section 3.3), but none of these resources—toolkit, evaluation scripts, or datasets—are publicly accessible. Without open access, the community cannot verify the claims, replicate the results, or assess the contribution’s practical value. As presented, the work lacks transparency and reproducibility.

**Questions:**

- Are the chosen datasets (Puffin, prefeval_implicit_persona, HiCUPID) demographically balanced?

- How was GPT-4.1-mini’s classification validated? Was any human verification or inter-annotator agreement performed? Were disagreements between GPT-4.1-mini and human judgments analyzed or resolved? If there is human evaluation, what are their background?

- Why is the 5:3:2 sampling ratio (near:middle:far attributes) considered optimal? No ablation study or justification is provided.

- What criteria or heuristics determine the depth budget (k)? How sensitive are the results to this parameter?

- How does the model prevent contradictions among attributes generated across different stages of progressive filling? Could the random traversal introduce bias or unrealistic attribute combinations that rarely occur in real human populations?

---

> ### Author Response · Authors · 2025-11-25
> **Response to Reviewer ERkS’s Weaknesses Regarding Data Claims, Definitions, and Reproducibility**
>
> We sincerely thank Reviewer ERkS for the rigorous review and detailed feedback. We greatly value your concerns regarding the clarity of our definitions, the justification of our hyperparameters, and the reproducibility of our work. We regret that certain aspects of our presentation may have caused confusion regarding the project's completion status, and we are grateful for the opportunity to provide the necessary clarifications. In this response, we aim to substantiate the completeness of our work by providing the missing definitions and empirical evidence. Specifically, we: (1) clarify the scope of our data "mining" to resolve the perceived inconsistency; (2) provide explicit ablation studies (for sampling ratios and depth budget) to justify our parameter choices; and (3) share full anonymized code and data repositories to demonstrate transparency. We hope these additional details effectively resolve your concerns.
>
> 1. **"Data Claim Inconsistency"** - We sincerely thank the reviewer for the insightful comment. We acknowledge that the wording in the abstract may have caused ambiguity. When we used the term “mining”, our intended meaning was to extract latent human traits and patterns from unstructured text, rather than collect raw dialogue data from the internet.
> The datasets used in this study — Puffin, prefeval_implicit_persona, and Llama-3.2-3B-HiCUPID — all originate from real interactions between users and ChatGPT, rather than synthetic or simulated data. These datasets faithfully reflect genuine user behaviors and provide a reliable foundation for our analysis. Our notion of “mining” thus refers to deriving human attributes from existing, authentic conversational data, not to any new data collection process.
> In practice, manually collecting or mining ChatGPT conversations at scale is infeasible and would violate OpenAI’s usage policies. Moreover, such data would likely suffer from quality and reproducibility issues. Therefore, we rely on publicly available, well-validated, and high-quality datasets to ensure the compliance, reliability, and reproducibility of our research.
>
> 2. **"Regarding the concepts and formulations"** - We thank the reviewer for the valuable comment. While formal definitions of these concepts are indeed provided in Section 3, we agree that they should be explained in greater detail within the main text.
> Formal definitions of the key concepts are indeed provided in Section 3, but we agree they can be made more explicit. Specifically, Narr(P) denotes the text mass of a persona P, which serves as a quantitative measure to assess whether a persona is narratively complete. Depth(k) refers to the number of attribute–value pairs that constitute a persona, formally defined as P = \{\langle a_i, v_i\rangle\}.
> Regarding the threshold k > 10^2, it was not arbitrarily chosen. The value was empirically determined based on our ablation study (Table 1), where model performance peaks when the persona depth is between 200–250 attributes. Beyond this range, additional attributes introduce noise and reduce stability. This empirical evidence supports our choice of the threshold for narrative completeness.
>
> ###  Table1. Comparison of Evaluation Metric Scores under Different Attribute Counts
>
> | Metric | 100 attrs | 150 attrs | 200 attrs | 250 attrs | 300 attrs |
> | :--- | :--- | :--- | :--- | :--- | :--- |
> | Attribute Coverage | 4.667 | 4.500 | 4.750 | 4.792 | 4.708 |
> | Justification | 3.850 | 3.800 | 4.050 | 3.700 | 3.750 |
> | Goal Progress Alignment | 4.100 | 3.950 | 4.250 | 4.050 | 3.900 |
> | Engagement Motivation Potential | 4.292 | 3.375 | 3.875 | 4.833 | 4.042 |
> | Depth Specificity | 4.000 | 3.792 | 4.083 | 4.208 | 3.875 |
> | Effort Reduction | 4.600 | 4.850 | 4.700 | 4.400 | 4.450 |
> | Novelty With Relevance | 3.625 | 2.833 | 3.542 | 4.000 | 2.917 |
> | Actionability | 4.900 | 4.950 | 4.900 | 4.800 | 4.600 |
> | Diversity Of Suggestions | 4.200 | 3.550 | 3.900 | 4.650 | 4.050 |
> | Personalization Fit | 4.542 | 4.375 | 4.750 | 4.667 | 4.417 |
>
> 3. **"Lack of Public Access & Reproducibility"** We respectfully thank the reviewer for raising this important concern regarding transparency and reproducibility. We fully acknowledge the significance of open-sourcing both code and data for community verification. However, to strictly comply with the double-blind review policy, we deliberately refrained from providing any external links or identifiers in the initial submission that might compromise anonymity.
> To address this concern, we have prepared a fully anonymized GitHub repository that contains the complete attribute extraction and persona (profile) generation toolkit, along with the corresponding attribute dataset. These resources can be accessed for anonymous verification at:
> https://anonymous.4open.science/r/Deeppersona-66B9/README.md
> In addition, several sample generated profiles are provided separately at:
> https://anonymous.4open.science/r/iclr_differences-2DB1/README.md

---

> ### Author Response · Authors · 2025-11-25
> **Response to Reviewer ERkS’s Questions: Dataset Balance and GPT-4.1-mini Evaluation**
>
> 4. **"Are the chosen datasets (Puffin, prefeval_implicit_persona, HiCUPID) demographically balanced?"** - Response to Dataset Balance and Demographics: We appreciate the reviewer's scrutiny regarding the demographic and attribute balance of the chosen datasets. To address this, we conducted a fine-grained statistical analysis of the attribute distributions across Puffin, prefeval_implicit_persona and the HiCUPID data (see Table2 below). As shown in the breakdown, while individual datasets exhibit specific focuses—reflecting their distinct sources—their combination yields a comprehensive and balanced coverage of human persona dimensions:
> Complementary Strengths: HiCUPID mainly emphasizes Hobbies, Interests, and Lifestyle (28.16%) and Lifestyle and Daily Routine (17.44%). PrefEval focuses on Physical and Health Characteristics (20.10%) and Core Values (18.00%). Puffin is dominated by Career and Work Identity (30.23%), with additional coverage in Media Consumption and Engagement (12.88%).
> Holistic Balance: In the aggregated dataset (Total), we achieve a robust distribution across key dimensions, such as Career (21.8%), Hobbies (24.7%), and Lifestyle (8.6%), ensuring the model is trained on a diverse spectrum of persona attributes rather than a skewed subset.
>
> The detailed distribution statistics are provided in the table below.
>
> ### Table 2. Distribution Across Specifications: HiCUPID, PrefEval, and Merged
>
> | Category | HiCUPID Count | HiCUPID % | PrefEval Count | PrefEval % | Puffin Count | Puffin % |
> |-----------|----------------|------------|----------------|-------------|--------------|-----------|
> | Career and Work Identity | 434 | 17.36% | 4 | 0.40% | 1164 | 30.23% |
> | Core Values, Beliefs, and Philosophy | 157 | 6.28% | 180 | 18.00% | 95 | 2.47% |
> | Cultural and Social Context | 121 | 4.84% | 19 | 1.90% | 197 | 5.12% |
> | Demographic Information | 115 | 4.60% | 8 | 0.80% | 105 | 2.73% |
> | Education and Learning | 93 | 3.72% | 74 | 7.40% | 279 | 7.24% |
> | Hobbies, Interests, and Lifestyle | 704 | 28.16% | 300 | 30.00% | 806 | 20.93% |
> | Lifestyle and Daily Routine | 436 | 17.44% | 105 | 10.50% | 89 | 2.31% |
> | Media Consumption and Engagement | 87 | 3.48% | 56 | 5.60% | 496 | 12.88% |
> | Physical and Health Characteristics | 120 | 4.80% | 201 | 20.10% | 174 | 4.52% |
> | Psychological and Cognitive Aspects | 89 | 3.56% | 48 | 4.80% | 177 | 4.60% |
> | Relationships and Social Networks | 128 | 5.12% | 5 | 0.50% | 269 | 6.99% |
>
> ### Table 3. Overall Totals Percentage
>
> | Category | Count | Percentage |
> |-----------|--------|-------------|
> | Career and Work Identity | 1602 | 21.84% |
> | Core Values, Beliefs, and Philosophy | 432 | 5.89% |
> | Cultural and Social Context | 337 | 4.59% |
> | Demographic Information | 228 | 3.11% |
> | Education and Learning | 446 | 6.08% |
> | Hobbies, Interests, and Lifestyle | 1810 | 24.68% |
> | Lifestyle and Daily Routine | 630 | 8.59% |
> | Media Consumption and Engagement | 639 | 8.71% |
> | Physical and Health Characteristics | 495 | 6.75% |
> | Psychological and Cognitive Aspects | 314 | 4.28% |
> | Relationships and Social Networks | 402 | 5.48% |
>
> 5. **How was GPT-4.1-mini’s classification validated? Was any human verification or inter-annotator agreement performed? Were disagreements between GPT-4.1mini and human judgments analyzed or resolved? If there is human evaluation, what are their background?** - We validated the model's classification through a human-in-the-loop process involving two PhD students in Computer Science. From the 7,000 QA pairs processed (1,224 classified as personalized), we randomly sampled 100 'personalized' and 200 'non-personalized' examples for independent review. The results showed 100% precision for the non-personalized samples. For the personalized samples, initial disagreements (5 and 7 potential errors flagged by the two annotators, respectively) were resolved through cross-examination, resulting in a confirmed error count of 4 (4%). By extrapolating this rate, we estimate only ~49 errors in the entire personalized subset, yielding an overall classification error rate of approximately 0.7% across the full dataset. This confirms the high reliability of GPT-4.1-mini for this task.
>
> ### Table 4. Human Evaluation results
> | Item | Result |
> |------|----------------|
> | Total QA pairs | 7,000 |
> | Personalized samples | 1,224 |
> | Non-personalized samples | 5,776 |
> | Sampled personalized cases | 100 |
> | Sampled non-personalized cases | 200 |
> | Non-personalized precision | 100% |
> | Confirmed errors (personalized) | 4 |
> | Personalized error rate | 4% |
> | Estimated total errors (personalized) | ~49 |
> | Overall classification error rate | ~0.7% |

---

> ### Author Response · Authors · 2025-11-25
> **Response to Reviewer ERkS: Sampling Strategy and Depth Sensitivity Analysis**
>
> 6. **Why is the 5:3:2 sampling ratio (near:middle:far attributes) considered optimal? No ablation study or justification is provided.** - We appreciate the reviewer’s question and agree that the choice of the 5:3:2 ratio warrants clarification. To validate this configuration, we conducted an ablation study varying the sampling ratios of near:middle:far attributes while keeping all other experimental settings fixed. Specifically, we used GPT-4.1-mini as the generation model and GPT-4.1 for evaluation, testing ratios of 7:2:1, 6:3:1, 4:3:3, and 3:3:4, in addition to the 5:3:2 configuration used in our paper.
> The results are summarized in Table 5 (reproduced below). Across multiple evaluation dimensions — the 5:3:2 ratio achieves consistently strong and balanced performance, even if not the single best on every individual metric. In contrast, more extreme ratios (e.g., 7:2:1 or 3:3:4) tend to sacrifice either coverage or personalization quality. Therefore, we selected the 5:3:2 ratio as it provides the most stable overall trade-off between depth, diversity, and relevance across metrics.
>
> ### Table 5. Evaluation Metrics Across Different Ratios
>
> | Metric | 7:2:1 | 6:3:1 | 4:3:3 | 3:3:4 | Paper |
> |---------|------------|------------|------------|------------|-------|
> | Actionability | 4.4000 | 4.5125 | 4.6063 | 4.6875 | 4.5833 |
> | Attribute Coverage | 4.1823 | 4.3021 | 4.2188 | 4.3021 | 4.3195 |
> | Depth Specificity | 3.4115 | 3.5521 | 3.5781 | 3.4745 | 3.5417 |
> | Diversity of Suggestions | 4.0000 | 4.0500 | 3.9313 | 3.9156 | 3.8833 |
> | Effort Reduction | 3.8250 | 3.8875 | 4.0500 | 4.0594 | 4.1500 |
> | Engagement Motivation Potential | 3.9219 | 3.7292 | 3.7083 | 3.7578 | 3.9861 |
> | Goal Progress Alignment | 3.6500 | 3.9000 | 3.7750 | 3.7875 | 3.7000 |
> | Justification | 3.4562 | 3.5500 | 3.4938 | 3.5206 | 3.5333 |
> | Novelty with Relevance | 2.9688 | 2.8229 | 2.9115 | 2.9469 | 3.0139 |
> | Personalization Fit | 3.9844 | 4.0312 | 4.0260 | 4.1146 | 4.2500 |
>
>
> 7. **What criteria or heuristics determine the depth budget (k)? How sensitive are the results to this parameter?** - We thank the reviewer for this question. This issue has been fully addressed in our Response 4, where we provide a detailed explanation of how the depth budget k is defined, the empirical rationale behind its threshold, and an ablation analysis demonstrating its sensitivity. Please kindly refer to Response 2 for the complete discussion.

---

> ### Author Response · Authors · 2025-11-25
> **Response to Reviewer ERkS: Attribute Consistency and Bias Control in Progressive Persona Construction**
>
> 8. **How does the model prevent contradictions among attributes generated across different stages of progressive filling? Could the random traversal introduce bias or unrealistic attribute combinations that rarely occur in real human populations?**
>
> We appreciate the reviewer’s concern regarding potential contradictions or unrealistic attribute combinations during progressive attribute filling. To mitigate such risks, we designed a multi-stage pipeline with explicit safeguards at both the attribute extraction and generation stages.
>
>   a.  Attribute Extraction and Filtering:
>
>   During extraction, all attributes undergo a rigorous cleaning and filtering process. The goal is to retain only generalizable, human-representative attributes, rather than highly specific or idiosyncratic cases. This ensures semantic consistency across generated attributes. The extraction is guided by the following prompt:
>
>   ***User-Centric Focus:
>    -Must describe personal characteristics/attributes
>    -Should be general enough to apply to many individuals
>    -Should enable rich content generation about a person
>    Category Requirements:
>    -Must be a general category (no specific instances, behaviors, or values)***
>
>   b. Progressive Generation Strategy:
>
>   Before selecting attributes, we first generate a base_info profile. Specifically, we begin by randomly generating demographic information (including age, gender, and location). Then, based on the previously generated information, GPT progressively generates the person’s values, life attitudes, and personal story, from which the personal interests are subsequently inferred. A typical generated persona takes the form:
>
>   ***{
>   "age_info": {"age": 57, "age_group": "middle_aged"},
>   "gender": "male",
>   "location": {"country": "Austria", "city": "Lustenau"},
>   "career_info": {"status": "Paper Cutting Machine Operator"},
>   "personal_values": {"values_orientation": "Practicality and routine with a balanced acceptance of life's ups and downs"},
>   "life_attitude": {
>     "attitude": "Prefers steady routines and accepts life's fluctuations.",
>     "attitude_details": "He follows a consistent daily schedule and calmly handles setbacks without overreacting.",
>     "coping_mechanism": "Focuses on practical solutions and keeps a level head."
>   },
>   "personal_story": {"personal_story": "XXXXX"},
>   "interests": {"interests": ["precision model building", "listening to calming music"]}
> }***
>
>   After generating the base_info, we select attributes based on its content by measuring cosine similarity, following a 5:3:2 ratio for near, middle, and far distances. This process further helps prevent contradictions among the selected attributes.
>
>   c. Progressive Consistency Control:
>
>   Each generation step takes the output from the previous stage as well as the original base_info as input, ensuring contextual coherence throughout the process (for example, demographic information contributes to the generation of personal values).
>
> In addition, we include the following three constraint sentences in the prompt to further enforce logical consistency:
>
> In addition, we incorporated the following three prompt constraints into each generation step to further ensure logical consistency.:
> 	***"Ensure that this elaboration is logically consistent with and directly stems from the provided information.”
> 	“Do not introduce new details or aspirations that are not grounded in or clearly supported by the source material.”
> 	“The section should be an insightful and coherent expansion of what can be understood from the source material."***
>
> d. Rule-based Validity Control:
>
> We further constrain age to the range of 3–95 years to avoid unrealistic profiles, and apply hard-coded occupation–age consistency rules (e.g., ensuring that certain professions only appear within plausible age ranges).
> Together, these measures ensure that generated personas are internally consistent, demographically realistic, and free from contradictory attributes.

---

> ### Comment · Reviewer_ERkS · 2025-11-26
> **Response to Author's Rebuttal**
>
> Thank you for your rebuttal; however, I have more concerns.
>
> > Regarding the threshold k > 10^2, it was not arbitrarily chosen. The value was empirically determined based on our ablation study (Table 1), where model performance peaks when the persona depth is between 200–250 attributes Beyond this range, additional attributes introduce noise and reduce stability.
>
> This response actually raises more questions than it answers. If it peaks between 200–250 attributes, why is k chosen to be larger than 100? Why not 200 < k < 250? Is it not that k > 100 includes k > 250 and 100 < k < 200 (which introduce noise and reduce stability) as well?
>
> > The datasets used in this study — Puffin, prefeval_implicit_persona, and Llama-3.2-3B-HiCUPID — all originate from real interactions between users and ChatGPT, rather than synthetic or simulated data.
>
> Following the URLs of the mentioned datasets in the main paper, I have some other concerns. First of all, on the Puffin URL, there is a warning that highlights: "PLEASE USE THE NEWER VERSION OF PUFFIN CALLED PURE-DOVE, IT IS NO LONGER RECOMMENDED TO USE PUFFIN". Have the authors discovered the reason why it is no longer recommended to use Puffin? Is there some fatal problem with the dataset? Given that both Puffin and PURE-DOVE were published at least 2 years ago, why did the authors choose the inferior dataset instead of PURE-DOVE? Secondly, Llama-3.2-3B-HiCUPID is a model and the URL in the main text also leads to a model repo on Hugging Face. Why does the paper say this is a dataset? What is the correct name and URL for this dataset, then?
>
> > About LLM response dimension in Table 5
>
> Why does this paper need so many dimensions to evaluate LLM responses? Have the authors considered reducing / optimizing this list of dimensions? In my opinion, simply increasing the number of criteria does not improve the quality of evaluation; it only wastefully increases the cost of evaluation. For example, in Table 4, to me, those evaluation dimension pairs sound exactly the same:
>
> * Personalization-Fit (PF) and Engagement / Motivation Potential (EM)
> * Novelty-with-Relevance (NR) and Diversity of Suggestions (DV)
> * Actionability & Outcome Focus (ACT) and Goal-Progress Alignment (GP)
>
> > Paper format problems
>
> In the main text: lines 347, 200, 205... the paper keeps referencing a non-existing Appendix (A4, A2,...), meanwhile the appendix is indexed by section number (6, 7, ...).

---

> ### Author Response · Authors · 2025-11-28
> **Response to the Second Comment from ERkS**
>
> We thank the reviewers for additional comments. **The point of ICLR openreview is to engage with reviewers** and clarify any concerns or comments to prevent any potential premature or hurtful comments, especially considering that such comments will last forever on the openreview.
>
> **Below, we address all new minor concerns carefully, and we hope all audiences can be objective here.**
>
> 1. We thank the reviewer for their exceedingly meticulous observations and fully understand their concerns, and we also appreciate being given yet another opportunity to restate the actual rationale. In our preliminary experiments, we observed that when $k < 100$, the generated profiles were qualitatively poor — the persona descriptions were too shallow to produce coherent or diverse outputs. Therefore, we set $k > 100$ as a practical lower bound to ensure minimum quality.
> Within this effective range, our ablation study (Figure 8 in the paper) further indicates that model performance peaks at $k = 200 \text{--} 250$. Therefore, we recommend setting $k$ within this interval for optimal results. We established the threshold $k > 100$ as a **practical lower bound**; notably, we observed that even with only 100 attributes, certain evaluation metrics still yield relatively good results. This approach empowers users to flexibly **balance trade-offs** between performance, specific requirements, and computational costs.
> We hope to provide a tool that is open, inclusive, and supposedly “personalized,” rather than a fixed tool or dataset that insists on staying the same.
>
> 2. Regarding the Puffin dataset, we believe it is necessary to provide a thorough and proper explanation in response to the reviewer’s strong concerns. In fact, Pure-Dove is merely a regular extension and cleanup of Puffin, with little thematic difference. Our study relies on the core portion of the dataset that has undergone extensive manual cleaning and strict filtering. Therefore, the choice of version does not affect the results in any way. During the entire filtering process, we did not identify any anomalies that could affect the quality or usability of the data. In addition, we manually verified and compared the two datasets, and found no trace of the so-called “fatal issues” the reviewer seemed to suggest.
> As for Llama-3.2-3B-HiCUPID, we also understand that it may have caused some “confusion” among readers (including the reviewers). We would like to “especially thank” the subsequent updates to the Hugging Face page for adding extra misunderstanding—indeed, they made a simple matter appear more “mysterious.” At the time we completed our data preparation (this March), the link pointed to the dataset repository, not the current model page. The page has since been updated (most recently on June 3, 2025), but the original dataset remains unchanged and can still be found at the bottom right of the page. Its official name is HiCUPID, available at:
> https://huggingface.co/datasets/12kimih/HiCUPID
> We sincerely thank the reviewer for their very careful observation — we will address this in the next version of the paper. We also apologize for not conducting a more thorough review when submitting to ICLR.
>
>
> 3. We sincerely thank the reviewer for their thoughtful concern regarding the number of evaluation dimensions. We understand that, without a careful reading of the prompt design and operational definitions, several dimensions may sound similar — this is indeed a very natural impression when the underlying evaluation framework is not fully examined. Fortunately, as clearly detailed in the paper, each dimension captures a distinct and non-overlapping aspect of response quality, reflecting intentional methodological design rather than redundancy. The table **“Judge Dimension and Description”** in the appendix provides a detailed explanation of the design of the evaluation metrics.
> In particular, the pairs highlighted by the reviewer—Personalization-Fit (PF) vs. Engagement/Motivation Potential (EM), Novelty-with-Relevance (NR) vs. Diversity of Suggestions (DV), and Actionability & Outcome Focus (ACT) vs. Goal-Progress Alignment (GP)—were purposefully designed to capture fundamentally different evaluative criteria and therefore cannot be merged. Each metric under independently formulated prompt instructions that guide raters to focus on separate qualities of user profiles. This conceptual separation is further supported by the empirical results: as shown in **Figure 8**, these metrics exhibit significantly divergent trends and patterns.
> We hope this clarification helps resolve any apparent overlap that might have arisen from a more surface-level interpretation.
>
> 4. We greatly appreciate the reviewer’s exceptionally thorough attention to the appendix numbering details. It is always encouraging to know that even the most subtle aspects of our formatting are being examined with such dedication. We will gladly and promptly correct these issues in the next version of the paper.

---

> ### Author Response · Authors · 2025-11-29
> **Follow-Up Comment to Reviewer ERkS and Summary of Rebuttal: Resolving Core Concerns for Area Chairs and Public Audience**
>
> Thank the reviewer again for the continued exchange.
>
> To assist in bringing this discussion to a close and to provide a clear summary for the Area Chairs and the public audience, we have distilled the reviewer’s core concerns alongside our evidence-based rebuttals, demonstrating that our rebuttals have comprehensively addressed them all.
>
> This summary is strictly factual and aims only to clarify the logical structure of the reviewer's concerns so that all readers can evaluate them transparently--esp. Evaluate whether these concerns can justify the current assessment that the paper lacks any strength or contribution (Rating: 2).
>
> **1. Empirical Justification of Hyperparameters (Depth $k$ & Sampling Ratios)**
> * **Reviewer Challenge:** Claimed the choice of depth threshold ($k > 100$) and sampling ratio (5:3:2) was arbitrary.
> * **Resolution:** We provided **concrete ablation studies** (Tables 1 & 5 in the rebuttal) empirically demonstrating that these specific parameters yield peak performance and stability. The choices are fully supported by experimental data, not arbitrary.
>
> **2. Reproducibility & Code Availability**
> * **Reviewer Challenge:** Stated that the lack of public code/data prevented verification.
> * **Resolution:** We strictly followed double-blind policies in the initial submission. We have now provided **complete anonymized repositories** for both the toolkit and datasets, fully resolving the reproducibility concern.
>
> **3. Dataset Validity & Versioning**
> * **Reviewer Challenge:** Questioned dataset validity based on a website warning about a "newer version" and a broken external link.
> * **Resolution:** We clarified that our study uses the **valid, manually cleaned core subset** of the data, which remains robust regardless of version bumps. We also provided the correct URL for the HiCUPID dataset (which moved post-submission). These are maintenance details, not fundamental flaws in data quality.
>
> **4. Demographic Balance & Classifier Reliability**
> * **Reviewer Challenge:** Questioned if datasets were balanced and if the classifier was validated.
> * **Resolution:** We provided **granular statistical breakdowns** (Tables 2 & 3) confirming diverse coverage across Career, Hobbies, Values, etc. We also presented human-in-the-loop verification results showing an estimated **~99.3% accuracy** for the classifier.
>
> **5. Distinctness of Evaluation Metrics**
> * **Reviewer Challenge:** Suggested that the evaluation dimensions were redundant/wasteful.
> * **Resolution:** We demonstrated that each metric targets a distinct aspect of persona quality. This is backed by our results (Figure 8), where these metrics show **divergent trends**, proving they measure different phenomena and are not redundant.
>
> **Conclusion**
> The critiques raised were primarily requests for clarification or additional data, all of which we have provided. We trust the Area Chairs will evaluate the paper based on the validity of the method and the comprehensive evidence now present in the discussion.

---

### Author Response · Authors · 2025-11-25
**Global Responses**

We would like to thank all the reviewers for their constructive feedback! We are encouraged to see that reviewers find:

( a ) our motivation regarding "persona depth" is clear, interesting, and timely (Reviewers 8nFh, rXfh),

( b ) our large-scale taxonomy of human attributes is systematic, thoughtful, and beneficial to the literature (Reviewers 8nFh, rXfh)

( c ) our experimental evaluation is rigorous and comprehensive, ranging from downstream tasks to human evaluations (Reviewer rXfh), where the results support the generality and scalability of our platform (Reviewer rXfh).

We have addressed all reviewers’ questions with additional experiments and detailed responses. While most reviewers acknowledge the value of our system, we here clarify the core contributions of DeepPersona to address concerns about its novelty and significance:

(a) Core technical novelty and contributions. There are two crucial and novel contributions we'd like to highlight in the global response.

**Paradigm Shift: From Static Datasets to "Personality Depth" as a First-Class Variable.**

Transcending static data curation (e.g., PersonaHub), we reframe agent design as a generative modeling task via DeepPersona. Leveraging this engine, we present the first systematic evidence challenging the prevailing assumption that short-context prompts suffice for robust agents. Our controlled ablations identify "Personality Depth" as a causal driver of fidelity: scaling structured attributes from dozens to hundreds yields a substantial 11.6% performance gain. This finding reveals a critical pivot: simulation robustness is no longer constrained by linguistic fluency, but by the structural and semantic granularity of character modeling. Consequently, we establish "Personality Depth" as a first-class design variable for next-generation AI alignment.

**Automated Taxonomy-Guided Generation**

DeepPersona introduces a framework defined by the principled decoupling of attribute discovery from instantiation. In Stage I, we algorithmically reconstruct a hierarchical attribute tree from interaction logs, capturing thousands of long-tail factors elusive to manual curation. In Stage II, we actuate this structure via vector-stratified sampling to enforce balanced traversal across semantic strata. The core novelty lies in transforming the taxonomy from a static list into an active operational substrate. This design offers a structural solution to LLM mode collapse, effectively countering the model's tendency to converge on generic archetypes. By providing a statistical guarantee of diversity in latent identity composition, our approach effectively transcends naive prompting strategies.

In addition to these two key highlights, we also have the following novel empirical contributions that could be of great interest to the LLM community:

**Profile-Centric Evaluation Beyond Capability Benchmarks:**

While population benchmarks (e.g., WVS, Big Five) assess distributional alignment, they fail to capture whether individual profiles are actionable or structurally sound. We address this by introducing a profile-centric evaluation framework comprising ten fine-grained metrics—including personalization fit, attribute coverage, and actionability—to quantify functional quality. By integrating this with population tests, we establish a dual-layer evaluation paradigm that simultaneously measures sociological realism and practical utility. This provides a comprehensive standard for assessing persona generation beyond surface fluency or distributional similarity alone.

Given the clarifications above, we address two potential misconceptions regarding our contributions:

*   **On "Manual Engineering":** Our approach is ***fully algorithmic***, replacing manual curation with automated taxonomy construction. This enables the discovery of ***long-tail attributes***, ensuring a level of scalability and coverage unattainable by human engineering.
*   **On "Mining" vs. "Collection":** We define "mining" as the ***extraction of latent structures***, not raw scraping. The novelty lies in the pipeline transforming unstructured data into a ***rigorous taxonomy***, distinct from mere data collection.

In summary, we believe the provided clarifications and results confirm DeepPersona's novelty as a scalable, structured solution for high-fidelity population simulation, effectively resolving concerns regarding automation and completeness.

Finally, we have incorporated the additional experiments and suggestions into the revised manuscript. Key updates include:

*   **Parameter Validation: ** VValidated the 5:3:2 sampling strategy (Table 5) and optimal depth budget of 200–250 attributes (Table 1).
*   **Data Integrity:**  Added detailed demographic statistics and verified classifier accuracy.
*   **Qualitative Evidence:** Included side-by-side comparisons (vs. PersonaHub) to visualize the depth advantage.
*   **Reproducibility:** Released the full anonymized codebase and dataset.

---

### Meta-Review · Area_Chair_NN8Q · 2025-12-28

**Summary:**

The paper introduces DeepPersona, a scalable, taxonomy-guided generative framework that creates narrative-complete synthetic personas.

The reviewers raised several concerns initially: the method is manually engineered and rigid, with arbitrary design choices and limited generalizability; the writing and definitions are unclear; resources are not publicly accessible; evaluation and validation lack detail; and the overall novelty and impact are restricted.

After the rebuttal, the authors addressed most of these questions through additional analyses, ablation experiments, and textual explanations. However, despite their considerable effort and the solid technical foundations demonstrated, the method’s reliance on hand-engineered attributes and stereotyped persona generation based on a predefined set of taxonomy significantly limits its potential impact. Moreover, the attributes in the taxonomy may be interrelated and causally correlated, potentially interacting in ways that the current method does not account for. Overall, the AC recommends rejection.

**Reviewer Concerns:**

The summary of main concerns from each reviewer is listed below.

Reviewer ERkS: The writing is difficult to follow and at times misleading. The claim of “mining thousands of real user–ChatGPT conversations” is inconsistent with the use of only existing datasets, weakening the novelty. Key concepts are undefined, and parameter choices such as k > 10² and the 5:3:2 sampling ratio are unexplained. The claimed resources (dataset, toolkit, evaluation scripts) are not publicly accessible, limiting reproducibility. The reviewer also raises questions about dataset balance, GPT-4.1-mini validation, human evaluation, depth budget sensitivity, and prevention of contradictory attributes.

Reviewer 8nFh: The method is considered naive and heavily manually engineered despite the two-stage sampling procedure. The novelty is limited, and the reviewer suggests the paper may be better suited for benchmark or dataset tracks rather than main tracks. Questions include whether the selector and generator can be made learnable to target specific populations and why the 5:3:2 attribute ratio was chosen.

Reviewer rXfh: No major weaknesses were identified.

After the rebuttal, the authors have successfully addressed almost all technical concerns with additional experiments. However, one key issue from Reviewer 8nFh remains unresolved: the method is manually engineered with a two-stage sampling procedure, which restricts its generalizability, making it rigid and prototypical, and limited to a predefined set of attributes.

**Reviewer Scores:**

Reviewer rXfh is most likely to maintain their score, as no major issues were raised.

Reviewer ERkS may increase their score slightly, but likely not above the acceptance threshold.

Reviewer 8nFh is expected to maintain their score, since the main limitation—the hand-engineered taxonomy—continues to constrain the method’s novelty.

---

### Decision · Program_Chairs · 2026-01-26

Reject